# NeRM: Learning Neural Representations for High-Framerate Human Motion Synthesis

**Dong Wei**[1], **Huaijiang Sun**[1]*, **Bin Li**[2], **Xiaoning Sun**[1], **Shengxiang Hu**[1], **Weiqing Li**[1], **Jianfeng Lu**[2]

[1]School of Computer Science and Engineering, Nanjing University of Science and Technology
[2]Tianjin AiForward Science and Technology Co., Ltd.
[1] `{csdwei,sunhuaijiang,sunxiaoning,hushengxiang,li_weiqing,lujf}@njust.edu.cn`
[2] `libin@aiforward.com`

## Abstract

Generating realistic human motions with high framerate is an underexplored task, due to the varied framerates of training data, huge memory burden brought by high framerates and slow sampling speed of generative models. Recent advances make a compromise for training by downsampling high-framerate details away and discarding low-framerate samples, which suffer from severe information loss and restricted-framerate generation. In this paper, we found that the recent emerging paradigm of Implicit Neural Representations (INRs) that encode a signal into a continuous function can effectively tackle this challenging problem. To this end, we introduce NeRM, a generative model capable of taking advantage of varied-size data and capturing variational distribution of motions for high-framerate motion synthesis. By optimizing latent representation and an auto-decoder conditioned on temporal coordinates, NeRM learns neural representations for sampled motion *clips* that ingeniously avoid explicit modeling of raw varied-size motions. This expressive latent representation is then used to learn a diffusion model that enables both unconditional and conditional generation of human motions. We show that NeRM not only achieves competitive results with state-of-the-art methods, but also be capable of generating arbitrary-framerate motions. Moreover, it can remain memory-friendly yet highly efficient even when generating high-framerate motions.

## 1 Introduction

The technology of human motion synthesis has been widely applied in various scenarios (Koppula & Saxena, 2013; Van Welbergen et al., 2010; Sun et al., 2023) of filming, robotics and gaming animation. In this task, generating high-quality motions with high framerate holds promising value for practical use (Feng et al., 2023; Wang et al., 2019). For example, the gaming system can showcase high-framerate animation on high-performance devices to enable a smooth and visually immersive gaming experience for players, while adaptively reducing the framerate on low-performance devices.

Existing methods (Petrovich et al., 2021; Zhang et al., 2022; Tevet et al., 2023; Cai et al., 2021; Wang et al., 2022) have neither considered, nor been able to handle such demands, which is primarily due to the following reasons: (1) High-framerate motion generation would impose substantial memory overloads that is unacceptable for current generative models, whose slow sampling speed further complicates this issue. (2) The raw motion data usually exhibits variability in framerates, such as text-to-motion dataset HumanML3D (Guo et al., 2022), wherein the framerate of each subset varies from 20 fps to 250 fps, making it challenging for direct use during training. As a result, all these methods employ a typical pre-processing step to construct a dataset with a fixed and target framerate. Motions exceeding the target framerate are downsampled, removing high-frequency details; and motions with inadequate framerate are excluded, discarding low-frequency structural information. However, such naïve process is inherently suboptimal, as it restricts the capabilities of data-driven models and can only generate motions with a fixed yet relatively low framerate. In practice, as human eyes are very sensitive to high-frequency details, it is desired to embrace the natural diversity

---
*Corresponding author.

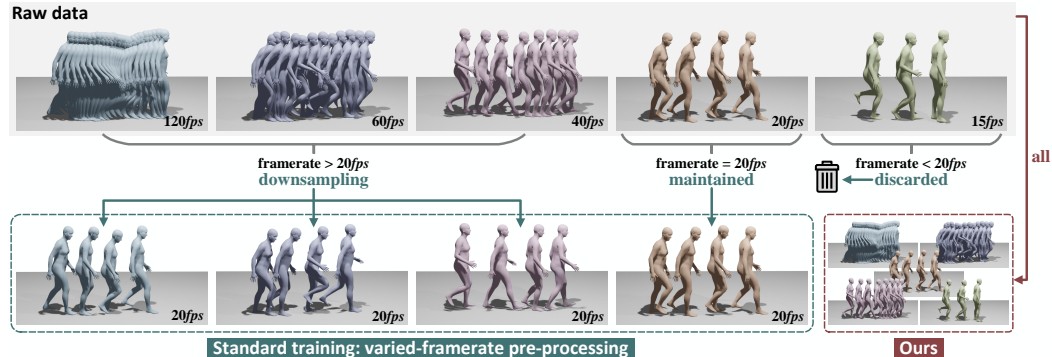

Figure 1: Motions are captured under different sampling rates. To realize uniform training on them as well as ensuring acceptable memory burden, existing models have to downsample sequences to a fixed, target framerate (such as 20 fps), and remove samples with that even lower. Our design can handle sequences at their native framerates, making full use of available annotated motion resources.

of motion framerates and process them at their *native* framerates, so that we can preserve and utilize the subtle nuances in original motions, thereby improving visual sensation for viewers.

Meanwhile, Implicit Neural Representations (INRs) (Mildenhall et al., 2021; Watson et al., 2023) have emerged as a new paradigm for representing continuous signals without explicit modeling, and gained popularity in various domains. For instance, Park et al. (2019) encodes geometry into a neural network that maps input 3D coordinates to corresponding SDF values. Accordingly, as opposed to discrete grid-wise signal values, INRs amortize the signal values of arbitrary coordinates into a compact neural representation, which eliminates the need for a large memory allocation that would be proportional to the coordinate dimension and resolution. In this respect, INRs have shown to be highly effective at modeling complex and dense signals, such as images (Skorokhodov et al., 2021), videos (Yu et al., 2022) and 3D scenes (Mildenhall et al., 2021).

Inspired by this, we propose NeRM, a variational implicit neural representation-based generative model for high-framerate human motion synthesis. NeRM represents a motion sequence as a continuous function $f : t \mapsto f(t)$ that is parametrized by a Multi-Layer Perceptron (MLP) conditioned on temporal coordinates. To mitigate excessive overfitting of motions, we introduce a latent code $z$, which can be thought of encoding of a sampled motion *clip* (a short segment of motion), as conditional variable to the function $f(t, z)$. When keeping the latent code $z$ constant and varying the coordinate $t$, NeRM allows us to efficiently generate motions at arbitrary framerates and time steps; and decouple high-framerate synthesis from prohibitive memory requirements. We find these latent codes by interpreting them as a variational distribution with optimized parameters (i.e., mean and covariance) in the representation space. This simple process naturally supports any-framerate training and showcases a significant departure from previous works (He et al., 2022; Mao et al., 2021; Wei et al., 2024) that require an explicit and sophisticated encoder to infer distributions of fixed-framerate motions. After optimizing the representative motion latent codes, motivated by the success of stable diffusion (Rombach et al., 2022), we learn to model the distribution of varied-framerate motions in latent space. Various conditions available to the model at training (e.g. action labels, text prompts) can be used to generate the realistic motions conforming to those conditions.

Additionally, acquiring high-quality human motion sequences with well annotations is expensive and limited (Chen et al., 2023). In this case, only temporal coordinate information is insufficient for the network to characterize continuous motion fields. We draw inspiration from Vector Quantized (VQ) codebook in 3D scenes (Yin et al., 2022), and inject such prior descriptors into the coordinated-based network. With the help of the codebook, our approach has the potential to enrich the feature representation of each temporal coordinate and then enhance the quality of generated motions.

We conduct comprehensive experiments on various datasets, including HumanML3D (Guo et al., 2022), KIT (Plappert et al., 2016), HumanAct12 (Guo et al., 2020) and UESTC (Ji et al., 2018). Numerical results demonstrate NeRM to be extremely competitive with state-of-the-art baselines. Furthermore, NeRM appreciates the following intriguing properties:

- *Arbitrary framerate training*: is capable of training on mixed-framerate datasets without pre-processing, like downsampling and discarding (Figure 1)

- *High-framerate motion generation*: can synthesize high-framerate motions (Figure 4b) of $\sim 120$ fps without suffering from increasing architecture size and massive memory resources (Appendix)

- *High inference speed*: achieve $120\times$ speed up in compared to Tevet et al. (2023) (Appendix)

- *Temporal sub-sampling*: enable the generation of poses at specific time steps directly without first generating frames before, while preserving the smoothness of the entire sequence (Figure 4c)

- *Flexible conditional generation*: allow for unconditional generation (Figure 5) as well as conditional on action labels (Table 3) or text descriptions (Table 1)

## 2 RELATED WORK

**Denoising Diffusion Probabilistic Models (DDPMs).** DDPMs (Sohl-Dickstein et al., 2015; Ho et al., 2020) have witnessed significant progress in the image synthesis domains, such as Imagen (Saharia et al., 2022) and DALL·E (Ramesh et al., 2022). Inspired by their works, most recent methods have adapted this advanced generative model to motion generation tasks. MotionDiffuse (Zhang et al., 2022) designs an effective DDPM-based architecture for controllable text-driven motion synthesis, allowing for motion manipulation in body parts. MDM (Tevet et al., 2023) makes predictions of the sample rather than the noise, facilitating the use of established geometric losses. As sampling of these approaches in raw motion space are computationally expensive, MLD (Chen et al., 2023) employs VAE with transformer backbone to map motions into latent space, where the diffusion model is trained. However, current motion diffusion models interpret the human motion as a sequence of poses, overlooking the inherent continuous temporal dynamics. They are also constrained to train exclusively on motions with a fixed yet relatively low framerate, which hinders their ability to generate motions at higher framerates. In this paper, we combines the merits of diffusion model and implicit neural representation to produce high-quality motions free from size limitations.

**Implicit Neural Representations (INRs).** INRs have shown extraordinary capability in modeling 3D shape neural representations (Park et al., 2019; Mescheder et al., 2019). In particular, NeRF (Mildenhall et al., 2021) uses multi-layer perceptron network to render 3D-consistent images with texture details. Following the success of NeRF, INRs have proven to be a powerful tool in many tasks such as view synthesis (Bautista et al., 2022; Watson et al., 2023), image generation (Dupont et al., 2023; Chai et al., 2022) and video encoding (Chen et al., 2021). Most recently, concurrent works (He et al., 2022; Cervantes et al., 2022; Wang et al., 2023) have been proposed to learn implicit neural fields to represent human motions. NeMF (He et al., 2022) adopts a VAE framework to make it a generative model controlled by latent code. However, this approach requires re-training the neural network to overfit a new signal, which is computationally costly. Moreover, NeMF can only find one possible latent code to satisfy conditional constraints, leading to deterministic motion generation like NeMo (Wang et al., 2023). Such problems have been addressed by (Cervantes et al., 2022), which generates novel motion sequences for a target action class by fitting conditional Gaussian Mixture Model (GMM). Unfortunately, this specific design can be only used for action-conditional motion generation, and is not versatile enough for other conditions (e.g. texts). In addition, these methods still do not support arbitrary-framerate training. In our approach, we draw upon the innovations in coordinate-based functions to sample motion *clips* at different framerates, and introduce the diffusion model to capture variational distribution of latents, thereby enhancing the capability of diverse motion sampling with flexible framerates and multimodal conditions.

## 3 METHOD

An overview of our NeRM is described in Figure 2. Our goal is to learn a generative model to synthesize a human motion given an arbitrary condition. Let $X = \{x^i\}_{i=1}^n$ denote a collection of training samples, where each sample $x^i = \{x_t^i\}_{t=1}^T$ is a motion sequence of $T$ human poses.

We decompose the task of learning a generative model in two stages. The first stage is responsible for capturing the motion of different framerates and durations, thereby learning a latent code $z_i$ for

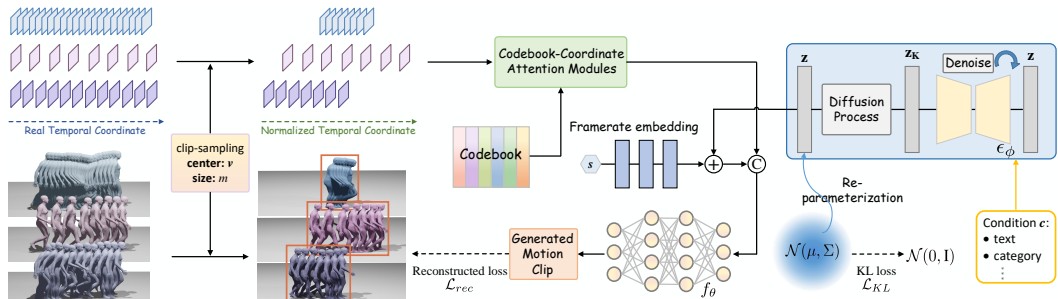

Figure 2: Two-stage pipeline of NeRM. In the first stage (left), we sample clips at random framerates from full-size motion sequences for training. A latent code $z$ is introduced as the encoding of these motion clips, which is directly optimized by the parameters of a posterior normal distribution instead of from a motion encoder. The latent, together with the codebook-enhanced normalized coordinates $t_{v,s}$ and the framerate $s$, are then fed into the decoder $f_\theta$ to produce motion clips. The second stage (right) uses the latent codes as input to our diffusion model and can be guided by various conditions.

each $x^i \in X$, while the second stage learns the distribution of the latent variables obtained from the first stage. This decouples the modeling of the complex distribution from varied-size motions.

## 3.1 MULTI-FRAMERATE VARIATIONAL IMPLICIT NEURAL REPRESENTATION

In standard motion synthesis, all training samples are typically required to have a consistent fixed framerate (see Figure 1). Our key idea is to naturally exploit the variety of motion framerates available in the dataset, learning from poses that are discarded, to enable high-framerate generation.

**Variational INRs.** Unlike most works that represent motion as a discrete sequential process (Wei et al., 2023), we represent motion as a continuous field of human poses in the temporal domain so that the inherent continuous temporal dynamics can be preserved. Formally, we denote a human pose of sequence $i$ at timestep $t$ as $x_t^i \in \mathbb{R}^{J \times D}$ that represented by either joint rotations or positions, where $J$ is the number of joints and $D$ is the dimension of the joint representation.

For each sequence $i$, we define the continuous motion field as a function $f_i(t) = \hat{x}_t^i$ that maps a temporal coordinate to a pose. By minimizing the reconstruction loss between ground truth $\{x_t^i\}_{t=1}^T$ and the generated motion $\{\hat{x}_t^i\}_{t=1}^T$, we can obtain an overfitting neural field $f_i(\cdot)$ for sequence $i$, thereby generating poses at arbitrary time steps by sampling the field. However, such modeling requires to be re-trained from scratch for a new motion sequence. Therefore, we introduce a latent code $z_i$ as an encoding of sequence $i$, thus parameterizing the whole training set $X$ as a function $f_\theta : (t, z_i) \mapsto \hat{x}_t^i$, where $f_\theta$ is a decoder shared among all sequences ($i \in \{1, 2, \cdots, n\}$), and $z_i$ is shared among all time steps ($t \in \{1, 2, \cdots, T\}$). This formulation allows us to swap out different latents $z$ for producing different motion sequences.

Nevertheless, the latent space distribution $p(Z)$ is not continuous since each $z_i$ is optimized independently to reconstruct a single sample via an over-parameterized decoder $f_\theta$. This renders the interpolation between different points in the latent space meaningless. Inspired by VAE (Kingma & Welling, 2013), we propose a variational INR to leverage the continuous nature of the latent space distribution. Specifically, we treat each latent code $z_i$ as a normal distribution, where the mean $\mu_i$ and covariance matrix $\Sigma_i$ are optimized. During training, we use the re-parameterization trick to sample an instance from this distribution. As such, the generated outputs exhibit smooth transitions between the corresponding $z_i$ and $z_j$. The motion field can thus be formulated as

$$f_\theta : (t, z_i) \mapsto \hat{x}_t^i, \quad \text{s.t.} \quad z_i \sim \mathcal{N}(\mu_i, \Sigma_i). \tag{1}$$

We represent human motions at arbitrary framerates $s$, with different durations $l$, as continuous functions over normalized temporal coordinates, and sample a random motion clip according to the center $v$ and the number of poses $m$ in clip. This enables any-framerate training of NeRM.

**Continuous-framerate training.** To learn from multi-framerate motions in training sets, we design our approach by exploiting the temporal consistency in motions to generate motion *clips*, which

is similar to patches in image. Images are continuous in 2D space, and motion sequences are continuous only in temporal dimension, but we can not simply handle this difference by ignoring one dimension. The essential difference between image and motion lies in that, image size is determined only by resolution, while motion size is determined by both framerate and duration.

For convenience, we assume a maximum duration of $l_{max}$ seconds for the motion and treat each motion as a continuous function defined on a bounded, normalized coordinate domain ranging from $-1$ to $1$. This implies that each second corresponds to an interval of $2/l_{max}$. The decoder $f_\theta$ always generates motion clips of a fixed number of poses $m$, but each clip implicitly corresponds to a short segment, centered at $v \in [-1, 1]$, of the longer (entire) motion. Denoting the framerate of the $i$-th entire motion as $s^i$ fps, we have that the clip size is $\frac{2}{l_{max}} \cdot \frac{m}{s^i}$ in normalized coordinates (see Figure 3). During training, we first sample clips from motions, and then pass them to the fixed-framerate decoder $f_\theta$. Therefore, Eqn.(1) can be reformulated as:

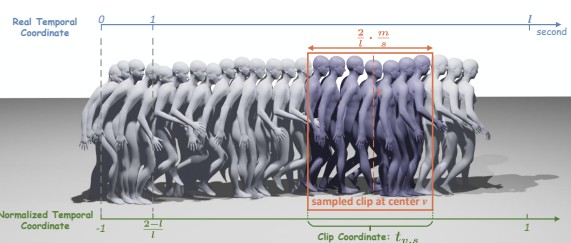

Figure 3: Illustrative description of the random clip sampling from the entire motion at framerate $s$ according to the center $v$ and the clip size $m$.

$$f_\theta : (t^i_{v,s}, \mathbf{z}_i, s^i) \mapsto \hat{x}^i_{clip}, \quad \text{s.t.} \quad \mathbf{z}_i \sim \mathcal{N}(\mu_i, \Sigma_i) \tag{2}$$

where $t^i_{v,s} \in \mathbb{R}^m$ denotes normalized temporal coordinates of motion clip with $m$ poses, which is extracted from all coordinates $t^i_s = [-1, \frac{-s^i \cdot l_{max}+2}{s^i \cdot l_{max}}, \frac{-s^i \cdot l_{max}+4}{s^i \cdot l_{max}}, \cdots, \frac{s^i \cdot l_{max}-2}{s^i \cdot l_{max}}, 1] \in \mathbb{R}^{l_{max} \cdot s^i}$ according to the clip center $v$, and $\hat{x}^i_{clip}$ denotes the corresponding human poses. Compared to current motion generative models (Tevet et al., 2023; Chen et al., 2023), our approach replaces the explicit input sequences with *clip-dependent coordinates* to allow for variable-framerate training.

**Codebook-enhanced representation.** As observed in Tancik et al. (2020), the Fourier embedding of coordinates endows INRs with the capability to learn high-frequency variations, addressing the problem know as "spectral bias" (Rahaman et al., 2019). Specifically, we employ $\gamma(t) = [\cos(2\pi b_1 t), \sin(2\pi b_1 t), \cdots, \cos(2\pi b_R t), \sin(2\pi b_R t)]^T$ as our embedding function, where $b_r$ is sampled from an isotropic distribution. However, we empirically find that such embedding is insufficient for the decoder $f_\theta$ to characterize human motion fields effectively.

To overcome this limitation, inspired by CoCo-NeRF (Yin et al., 2022) that utilizes codebook information for 3D geometry neural representations, we propose to deploy Codebook-Coordinate Attention (CCA) modulation to enrich the Fourier features of each coordinate. The detailed design of CCA can be found in Appendix. Given a pre-trained codebook containing $N$ codes $\mathcal{E} = \{e_i\}_{i=1}^N$ with $e_i \in \mathbb{R}^d$ where $d$ is the dimension of codes, we define $\mathcal{Q} = \{q_i\}_{i=1}^M$ as the learnable query vectors, which are used to query the motion-relevant prototypes from the codebook via a cross-attention mechanism. The feature representations of these prototypes are then iteratively boosted by passing several self-attention layers. Subsequently, the prototype information is gradually incorporated into each $\gamma(t)$ through cross-attention modules. This workflow establishes a connection between the codebook prior and coordinate embedding, thereby improving its feature representation.

**Loss objective.** For the motion sequence $i$, NeRM calculates the reconstruction loss over all time steps of the generated motion clip and the ground truth, and uses the Kullback-Leibler (KL) divergence to encourage the generated latent space to approach a target distribution:

$$\mathcal{L}^i = \mathcal{L}^i_{rec} + \lambda_{KL}\mathcal{L}^i_{KL} = \|\hat{x}^i_{clip} - x^i_{clip}\|^2 + \lambda_{KL}D_{\text{KL}}(\mathcal{N}(\mu_i, \Sigma_i)\|p(z)), \tag{3}$$

where we regularize the posterior to match a prior $p(z)$ that is set as $\mathcal{N}(0, \mathbf{I})$. Diffusion processes (which we show later) converge towards Gaussian distributions so modeling data to approximate this distribution results in faster and more stable training. We also add $\lambda_{KL}$ to control the strength of regularization. The optimization problem for the model parameters can then be defined as following:

$$\{(\mu^*_i, \Sigma^*_i)\}_{i=1,2,\cdots,n} = \underset{\mu_i, \Sigma_i}{\arg\min} \mathcal{L}^i, \qquad \theta^* = \underset{\theta}{\arg\min} \sum_{i=1}^n \underset{\mu_i, \Sigma_i}{\min} \mathcal{L}^i, \tag{4}$$

where the first term represents the clip-wise parameters for each sequence, while the second term represents the decoder parameters that are shared across the entire dataset, with multiple framerates.

**Progressive training.** At the beginning of multi-framerate training, we first train at a fixed framerate to learn globally coherent motions, like conventional motion synthesis methods. The clip temporal coordinate $t_{v,s}$ and the framerate $s$ are unchanged. Due to the lower computational cost at this time, we are able to train the model for an extended number of iterations associated with this setup. Subsequently, we involves multiple framerates. As the model has already learned crucial representations from fixed-framerate training, fewer iterations are required to achieve convergence.

## 3.2 CONDITIONAL MOTION LATENT DIFFUSION MODEL

Given a set of latent codes $Z = \{z_i \sim \mathcal{N}(\mu_i^*, \Sigma_i^*)\}_{i=1}^n$ optimized in (4), we aim at learning Denoising Diffusion Probabilistic Models (DDPMs) (Ho et al., 2020) to capture the distribution $p(Z)$ in their latent space. In general, DDPMs with discrete timesteps have a fixed Markovian forward process $q(z_k|z_{k-1})$ where $q(z_0)$ denotes the data distribution and $q(z_K)$ is defined to approximate the standard normal distribution, where the subscript represents the time step. DDPMs then learn to reverse the forward process $p_\phi(z_{k-1}|z_k)$ with learnable parameters $\phi$. We train our latents $p_\phi(Z)$ by learning to denoise $z_k$ to $z_0$ for all timesteps $k$ (Ho et al., 2020), which can be formulated as:

$$\min_\phi \mathbb{E}_{k,z \sim Z, \epsilon \sim \mathcal{N}(0,\mathrm{I})} \left[ \| \epsilon - \epsilon_\phi(\sqrt{\bar{\alpha}_k} z + \sqrt{1 - \bar{\alpha}_k} \epsilon, k) \|^2 \right], \tag{5}$$

where $k$ denotes the timestep sampled from a uniform distribution, $\bar{\alpha}_k$ denotes a noise magnitude parameter with a fixed scheduling, $\epsilon \in \mathcal{N}(0, \mathrm{I})$ is the noise, and $\epsilon_\phi$ denotes the learned denoising model. We build a transformer-based denoising model with long skip connections for $\epsilon_\phi$, whose superiority has already been proved by time series data (Bao et al., 2023).

**Flexible conditions.** Our diffusion model in latent space also supports conditional motion generation by capturing distribution of $p(Z|C)$, such as text and action. Given paired data $\{z \in Z, c \in C\}$, only the denoising model $\epsilon_\phi$ need to be augmented with a conditioning variable $c$ to form $\epsilon_\phi(z, k, c)$, sharing a common motion variational INRs. To address various $c$, the domain encoder is employed for condition embedding, e.g., we employ the frozen text encoder of CLIP (Radford et al., 2021) to map text prompt, and build the learnable embedding (Petrovich et al., 2021) for each action category. We incorporate these embedded conditions into a transformer-based $\epsilon_\phi$ by concatenation.

**Sampling.** At the inference time, we start by sampling $z_K \sim \mathcal{N}(0, \mathrm{I})$ and iteratively apply $\epsilon_\phi$ to denoise $z_K$. We also perform classifier-free guidance (Ho & Salimans, 2022) with conditional variable $c$ to increase diversity and prevent overfitting. In practice, we use a zero-mask instead of the condition with a certain probability at each training iteration. When sampling, we have:

$$\epsilon_\phi(z_k, k, c) = r \epsilon_\phi(z_k, k, c) + (1 - r) \epsilon_\theta(z_k, k, \emptyset), \tag{6}$$

where $r$ is the guidance scale. After reversing the diffusion Markov Chain to obtain $z_0$, we feed it as input to the variational INR decoder $f_\theta$ and reconstructs plausible human motions.

## 4 EXPERIMENTS

We evaluate NeRM on tasks of: (1) text-to-motion, (2) action-to-motion and (3) unconditional motion generation. We provide dataset introduction, evaluation metrics, results and visualizations. We present a new motion quality metric *clip-FID* that measures FID over varied-framerate motions, and quantifies the performance of our distinctive high-framerate property. More qualitative results, ablation studies and implementation details are in Appendix.

### 4.1 TEXT-TO-MOTION

**Datasets.** Given free-form texts, we conduct text-to-motion experiments on KIT Motion-Language dataset (Plappert et al., 2016) and HumanML3D (Guo et al., 2022). (1) The former one is composed of 3,911 human motion sequences with 6,353 free-form texts as descriptions. Usually, prior works (Tevet et al., 2023) downsample the 100 fps motion data into 12.5 fps for training and testing; we, however, *maintain its original framerate*. (2) The latter one is a recently proposed dataset that integrates AMASS (Mahmood et al., 2019) and HumanAct12 (Guo et al., 2020) to form 14,616 motion

Table 1: Results of conventional text-to-motion synthesis on HumanML3D and KIT dataset. All methods use the real motion length from the ground truth for guidance. The right arrow → means results are better when closer to that of real motion. - means unavailable results. **Bold** indicates best results; underline indicates second best; ± indicates 95% confidence interval.

| Method | HumanML3D (Guo et al., 2022) | | | | | KIT (Plappert et al., 2016) | | | | |
|---|---|---|---|---|---|---|---|---|---|---|
| | FID ↓ | R-Precision (Top-3) ↑ | Multimodal Dist ↓ | Diversity → | MM ↑ | FID ↓ | R-Precision (Top-3) ↑ | Multimodal Dist ↓ | Diversity → | MM ↑ |
| Real | $0.002^{\pm.000}$ | $0.797^{\pm.002}$ | $2.974^{\pm.008}$ | $9.503^{\pm.065}$ | - | $0.031^{\pm.004}$ | $0.779^{\pm.006}$ | $2.788^{\pm.012}$ | $11.08^{\pm.097}$ | - |
| JL2P (Ahuja & Morency, 2019) | $11.02^{\pm.046}$ | $0.486^{\pm.002}$ | $5.296^{\pm.008}$ | $7.676^{\pm.058}$ | - | $6.545^{\pm.072}$ | $0.483^{\pm.005}$ | $5.147^{\pm.030}$ | $9.073^{\pm.100}$ | - |
| Hier (Ghosh et al., 2021) | $6.532^{\pm.024}$ | $0.552^{\pm.004}$ | $5.012^{\pm.018}$ | $8.332^{\pm.042}$ | - | $5.203^{\pm.107}$ | $0.531^{\pm.007}$ | $4.986^{\pm.027}$ | $9.563^{\pm.072}$ | - |
| T2M (Guo et al., 2022) | $1.067^{\pm.002}$ | $0.740^{\pm.003}$ | $3.340^{\pm.008}$ | $9.188^{\pm.002}$ | $2.090^{\pm.083}$ | $2.770^{\pm.109}$ | $0.693^{\pm.007}$ | $3.401^{\pm.008}$ | $10.91^{\pm.119}$ | $1.482^{\pm.065}$ |
| MoFusion (Dabral et al., 2023) | - | $0.492$ | - | $8.82$ | $2.521$ | - | - | - | - | - |
| MDM (Tevet et al., 2023) | $0.544^{\pm.044}$ | $0.611^{\pm.007}$ | $5.566^{\pm.027}$ | $9.559^{\pm.086}$ | **$2.799^{\pm.072}$** | $0.497^{\pm.021}$ | $0.396^{\pm.004}$ | $9.191^{\pm.022}$ | $10.85^{\pm.109}$ | $1.907^{\pm.214}$ |
| PhysDiff (Yuan et al., 2023) | $0.433$ | $0.631$ | - | - | - | - | - | - | - | - |
| MLD (Chen et al., 2023) | $0.473^{\pm.013}$ | $0.772^{\pm.002}$ | $3.196^{\pm.010}$ | $9.724^{\pm.082}$ | $2.413^{\pm.079}$ | **$0.404^{\pm.027}$** | $0.734^{\pm.007}$ | $3.204^{\pm.027}$ | $10.80^{\pm.117}$ | **$2.192^{\pm.071}$** |
| NeRM (fixed-framerate train) | $0.489^{\pm.013}$ | $0.774^{\pm.003}$ | $3.186^{\pm.015}$ | $9.692^{\pm.082}$ | $2.330^{\pm.075}$ | $0.522^{\pm.029}$ | $0.727^{\pm.005}$ | $3.843^{\pm.027}$ | $10.90^{\pm.094}$ | $1.931^{\pm.133}$ |
| NeRM (native-framerate train) | **$0.389^{\pm.011}$** | **$0.779^{\pm.003}$** | **$3.178^{\pm.016}$** | **$9.547^{\pm.073}$** | $2.193^{\pm.081}$ | $0.472^{\pm.019}$ | **$0.736^{\pm.007}$** | **$3.189^{\pm.031}$** | **$10.94^{\pm.114}$** | $1.785^{\pm.082}$ |

sequences with 44,970 textual annotations. It totally involves 19 sub-datasets, whose framerates vary from 20 fps to 250 fps. Existing motion generative models downsample them into 20 fps, and represent each frame by concatenating root velocities, joint positions, joint velocities, joint rotations and foot contact binary labels. We keep this representation, but *on their original framerates*.

**Evaluation metrics.** Following Guo et al. (2022), we evaluate our model by calculating: (1) Frechet Inception Distance (FID), the similarity between the distribution of generations and real motions; (2) R-Precision and Multimodal-Dist, the degree to which the generated motions adhere to the given texts; (3) Diversity, the variance within the generated distribution; (4) Multimodality (MM), the generation diversity within a single text prompt. In addition, the aforementioned metrics are designed for evaluating global structure, but ignore the realism of details. We thus propose a new metric dubbed (5) **clip-FID** to evaluate the quality of high-framerate generative details. It randomly extracts small clips from high-framerate real/generated motions, and calculates FID over ground truth and generated motion clips, sampled at random center $v$ and clip size $m$. As clip-FID preserves the target framerates without downsampling, we find it more sensitive to local details, enabling a more accurate assessment of artifacts such as foot sliding. Detailed introductions are in Appendix.

**Conventional motion generation.** We compare our NeRM to text-to-motion baselines JL2P (Ahuja & Morency, 2019), Hier (Ghosh et al., 2021), T2M (Guo et al., 2022), and more recent motion diffusion models MDM (Tevet et al., 2023), MoFusion (Dabral et al., 2023), PhysDiff (Yuan et al., 2023), and MLD (Chen et al., 2023). In Table 1, we show our performance of 20 fps generation on HumanML3D and 12.5 fps on KIT, which are aligned with the generations of existing baselines. Our method outperforms these baselines in terms of FID, R-Precision, Multimodal Dist and Diversity on both datasets, which proves that utilizing the native framerates of raw data is much beneficial for generation. This advantage exactly stems from our capability of handling varied-framerate data that other models are unable to process. It is worth mentioning that our method achieves better results on HumanML3D dataset than on KIT dataset because the former one has more abundant motion data.

Notably, since NeRM is constructed with two stages (INR + diffusion), we draw our attention to the comparison with MLD (Chen et al., 2023) that also generates motions with a two-stage (VAE + diffusion) design. Our model outperforms MLD on both datasets. This is owing to the advantage of continuous motion fields in INR-based framework, rather than explicitly modeling motion inputs. Additionally, compared to transformer backbone in VAE decoder of MLD, NeRM achieves fast inference speed due to its simple MLP decoder of INR, which can be found in Appendix.

**Fairness discussion.** To remove our superiority brought by raw training data, we also *train NeRM on pre-processed fixed-framerate datasets* for fair comparison with baselines. From Table 1, NeRM (fixed-framerate train) obtains comparable results to others, which can be attributed to our powerful latent diffusion framework and codebook-enhanced representation. Also, it is significantly surpassed by our native-framerate training, which further confirms the effectiveness of using raw data.

**High-framerate motion generation.** Results in Table 2 show our unique property of generating high-quality motions at different high framerates. However, for baselines, they can neither undertake the memory burden and sampling time brought by high-framerate motions, nor effectively train on

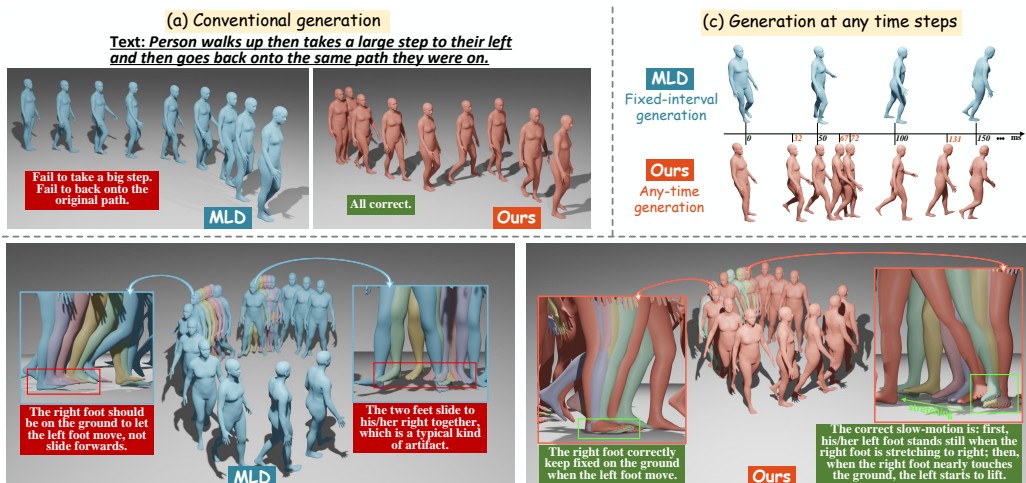

Figure 4: (a) Generating 20 fps motion. The failure in MLD is removed by NeRM. (b) Generating 100 fps motion, where high-framerate details are highlighted with purple, pink, yellow and green sequentially. NeRM directly generates them, showing delicate, realistic and reasonable movement details; while MLD can only realize low-framerate generation, so we attach it with **interpolation** to force it into high-framerate, which appears foot sliding artifacts. (c) Temporal sub-sampling: NeRM can realize any-time generation, while current models (e.g. MLD) cannot.

varied-framerate raw data. Therefore, for fair comparison under high-framerate setting, we conduct upsampling by spherical linear interpolation (Shoemake, 1985) on their low-framerate generations towards the target framerate, and calculate the corresponding clip-FID values. NeRM yields much better performance on all the target framerates, indicated by our clip-FID metric that can provide special evaluation on generated high-framerate details.

**Visualizations.** We further investigate the ability of NeRM by the qualitative analysis. Figure 4a illustrates the visualizations of conventional 20 fps generation on a sample from HumanML3D between MLD and ours. The failure of MLD does not happen to our model, which proves that our full usage of raw data and continuous human motion

Table 2: Evaluation of generated motions at different framerate (fps) on HumanML3D dataset using clip-FID.

| Method | 20 | 40 | 60 | 100 | 120 |
|---|---|---|---|---|---|
| T2M | 1.067 | 2.831 | 6.442 | 9.182 | 11.264 |
| MDM | 0.544 | 1.882 | 4.605 | 5.966 | 8.401 |
| MLD | 0.473 | 1.465 | 3.816 | 5.138 | 7.878 |
| NeRM | **0.389** | **0.493** | **0.680** | **0.903** | **1.315** |

fields are effective for high-quality motion generation. To demonstrate the high-framerate generation ability of NeRM, we also exhibit their visualized comparison over high-framerate performances in Figure 4b. We set the target framerate to 100 fps and amplify the details on human feet. We observe that our model yields smooth and natural generations, while the baseline can only generate the fixed 20 fps and then conduct motion interpolation towards 100 fps. This operation will lead to a typical kind of low-quality phenomenon, i.e., foot sliding, which results in incoherent and less life-like human behaviors in high-performance devices. Remarkably, another unique property of our NeRM is to enable the generation of a pose at specific time step directly without first generating frames before, while maintaining the smoothness of the entire sequence. Existing state-of-the-art text-to-motion methods have to sequentially generate the motion at fixed time intervals, which limits their application to temporal sub-sampling. As shown in Figure 4c, NeRM is able to infer any frame by sampling the temporal field $t$.

## 4.2 ACTION-TO-MOTION

**Datasets & Metrics.** Action-to-motion aims at generating relevant motion sequences given an input action label. We experiment on UESTC (Ji et al., 2018) and HumanAct12 (Guo et al., 2020). The former one is composed of 40 action classes with 25K samples, while the latter contains 12 classes

Table 3: Quantitative results of action-to-motion synthesis on UESTC and HumanAct12 dataset.

| Method | UESTC (Ji et al., 2018) | | | | | HumanAct12 (Guo et al., 2020) | | | |
|---|---|---|---|---|---|---|---|---|---|
| | $\text{FID}_{train} \downarrow$ | $\text{FID}_{test} \downarrow$ | Accuracy $\uparrow$ | Diversity $\rightarrow$ | MM $\rightarrow$ | $\text{FID}_{train} \downarrow$ | Accuracy $\uparrow$ | Diversity $\rightarrow$ | MM $\rightarrow$ |
| Real | $2.92^{\pm.26}$ | $2.79^{\pm.29}$ | $0.988^{\pm.001}$ | $33.34^{\pm.320}$ | $14.16^{\pm.06}$ | $0.020^{\pm.010}$ | $0.997^{\pm.001}$ | $6.850^{\pm.050}$ | $2.450^{\pm.040}$ |
| ACTOR (Petrovich et al., 2021) | $20.5^{\pm2.3}$ | $23.43^{\pm2.20}$ | $0.911^{\pm.003}$ | $31.96^{\pm.33}$ | $14.52^{\pm.09}$ | $0.120^{\pm.000}$ | $0.955^{\pm.008}$ | $6.840^{\pm.030}$ | $2.530^{\pm.020}$ |
| MDM (Tevet et al., 2023) | $\underline{9.98}^{\pm1.33}$ | $\mathbf{12.81}^{\pm\mathbf{1.46}}$ | $0.950^{\pm.000}$ | $33.02^{\pm.28}$ | $\mathbf{14.26}^{\pm.\mathbf{12}}$ | $\underline{0.100}^{\pm.000}$ | $\mathbf{0.990}^{\pm.\mathbf{000}}$ | $6.680^{\pm.050}$ | $2.520^{\pm.010}$ |
| MLD (Chen et al., 2023) | $12.89^{\pm.109}$ | $15.79^{\pm.079}$ | $\underline{0.954}^{\pm.001}$ | $\underline{33.52}^{\pm.14}$ | $13.57^{\pm.06}$ | $\mathbf{0.077}^{\pm.\mathbf{004}}$ | $0.964^{\pm.002}$ | $\underline{6.831}^{\pm.050}$ | $2.824^{\pm.038}$ |
| INR-MLP (Cervantes et al., 2022) | $\mathbf{9.55}^{\pm.\mathbf{06}}$ | $15.00^{\pm.09}$ | $0.941^{\pm.001}$ | $31.59^{\pm.19}$ | $14.68^{\pm.07}$ | $0.114^{\pm.001}$ | $0.970^{\pm.001}$ | $6.786^{\pm.057}$ | $\underline{2.507}^{\pm.034}$ |
| NeRM (Ours) | $11.75^{\pm.31}$ | $\underline{14.23}^{\pm.174}$ | $\mathbf{0.956}^{\pm.\mathbf{001}}$ | $33.20^{\pm.21}$ | $14.41^{\pm.06}$ | $0.106^{\pm.000}$ | $\underline{0.977}^{\pm.001}$ | $\mathbf{6.866}^{\pm.\mathbf{032}}$ | $2.492^{\pm.048}$ |

with 1,191 samples. As these two datasets do not provide high-framerate data, we remain their original framerates during training, and evaluate NeRM using the set of metrics in (Guo et al., 2020), including FID, Accuracy, Diversity and Multimodality. This task indicates that our NeRM can accept other multi-modal signals as conditions.

**Results.** We compare our NeRM to baseline ACTOR (Petrovich et al., 2021), INR-MLP (Cervantes et al., 2022), MDM (Tevet et al., 2023) and MLD (Chen et al., 2023), where ACTOR and MLD are under transformer-based VAE setting; MDM and MLD under diffusion setting. As provided by Tevet et al. (2023), we perform 20 evaluations, each consisting of 1000 samples, and present the average along with a 95% confident interval. Quantitative comparison is reported in Table 3, where NeRM achieves competitive results on both datasets. It is worth mentioning that INR-MLP (Cervantes et al., 2022) is also built upon implicit neural representation and consist of MLP layers, similar to ours. However, INR-MLP can only accomplish the action-conditioned generation task and cannot apply to other tasks due to its specific design. Moreover, it focuses on variable-length motion generation and does not consider arbitrary-framerate training sets and high-framerate generation. NeRM outperforms INR-MLP in terms of Accuracy and Diversity, demonstrating the high capability of latent diffusion model and the enhanced representation of codebook-coordinate attention.

## 4.3 UNCONDITIONAL GENERATION

**Datasets & Metrics.** Finally, we use part of AMASS (Mahmood et al., 2019) dataset to evaluate the generations of NeRM. This dataset can be also regarded as the motion part of HumanML3D (Guo et al., 2022), discarding its text part. We train on its native varied framerates, with FID, clip-FID and Diversity for motion quality and diversity.

**Results.** We compare NeRM with baseline ACTOR (Petrovich et al., 2021), MDM (Tevet et al., 2023), NeMF He et al. (2022) and MLD (Chen et al., 2023). For AC-TOR, we use the 6-layers transformer VAE from it, and then follow TEMOS (Petrovich et al., 2022) to make it class-agnostic, as suggested by MLD. The other three methods naturally support this task. In Figure 5, we show that NeRM achieves the best performance against baselines over conventional generation in terms of motion quality, and competitive results in terms of diversity. NeRM also outperforms INR-based method NeMF due to

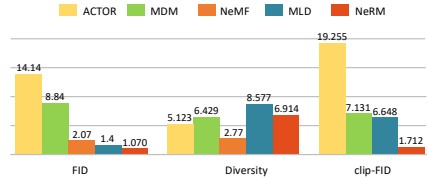

Figure 5: Comparison of unconditional motion generation on part of AMASS dataset with state-of-the-art methods.

flexible multi-framerate training and powerful generative ability of the diffusion model.

## 5 CONCLUSION

In this work, we present NeRM, a novel neural representation for human motions, which involves learning a latent representation and then capturing its distribution with diffusion models. Our key insight is that by directly learning a continuous motion field over temporal coordinates without explicit modeling, we can train a neural network jointly on processed low-framerate motions to learn global structure and on clips from the raw varied-framerate motions to learn details. This lifts the fixed-framerate requirement of previous motion generative models that treat motions as consecutive frame sequence and discard higher-frequency details. With such representation, we show that NeRM can efficiently generate high-quality motions at high framerates with flexible conditions controlled.

ACKNOWLEDGMENT

This work was supported in part by the National Natural Science Foundation of China (NO. 62176125, 61772272).

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

# A    OTHER RELATED WORK

**Human Motion Synthesis.** Human motion synthesis aims at generating natural motions consistent with any signal that describes the motion, such as action category (Guo et al., 2020; Petrovich et al., 2021), text (Guo et al., 2022; Tevet et al., 2023), music (Tseng et al., 2023; Alexanderson et al., 2023) and historical pose sequences (Mao et al., 2019; Li et al., 2020). With these guidance, earlier works (Ahuja & Morency, 2019; Ghosh et al., 2021) focus on learning a shared latent space for motion and conditions deterministically, limiting one-to-one mapping from condition to motion. Some recent works have put more emphasis on the promotion of diversity, and learned to model the distribution of motions based on the development of deep generative models, like Variational AutoEncoders (VAEs) (Kingma & Welling, 2013) and Generative Adversarial Networks (GANs) (Goodfellow et al., 2014). BiHMP-GAN (Kundu et al., 2019), conditioned on a given starting sequence, uses the discriminator of GANs to regress the random vector for multiple probable predictions. (Cai et al., 2021) presents a VAE-based unified framework for generalized motion synthesis that covers motion prediction, completion, interpolation and recovery. Wang et al. (2022) involves the modeling of human-scene interaction, path planning and body movement, to implement motion generation in the given scene environment with target action sequence. ACTOR (Petrovich et al., 2021) and TEMOS (Petrovich et al., 2022) suggest employing a VAE to map action labels and texts into a variational distribution with transformer structure (Vaswani et al., 2017), respectively.

However, all the aforementioned methods primarily focus on fixed-size motion modeling, and do not take into consideration the aspect of motion framerate. As a result, higher-framerate details and lower-framerate global structures in datasets are disregarded. Moreover, these methods are constrained to generating motions of fixed sizes. In contrast, our proposed method addresses these limitations by incorporating available varied framerates into considerations and enabling the generation of motions at arbitrary framerates.

**Motion Representation.** Motion representation is very crucial for the subsequent synthesis task. There existing several motion representation on different datasets, such as (1) the classical SMPL-based motion parameters (Petrovich et al., 2021), (2) the redundant hand-crafted motion features (Guo et al., 2022), and (3) straight-forward joint positions (Mao et al., 2019). We opt for the first two representations in this paper. Particularly, The first one is widely used in motion capture, and the second one is mainly used in character animation. Following (Tevet et al., 2023; Chen et al., 2023), we employ the SPML parameters in action-to-motion task for a fair comparison, and redundant hand-crafted features in text-to-motion and unconditional tasks.

# B    MORE DETAILS

## B.1    IMPLEMENTATION

Our NeRM are decomposed into two stages, including INR and latent diffusion. For INR, the hidden layer size is fixed to 1,024. We use the pre-trained codebook of (Zhang et al., 2023a), which is trained by VQ-VAE (Van Den Oord et al., 2017) on the HumanML3D dataset. The codebook size is set to $512 \times 512$. The number of learnable query embeddings of codebook-coordinate attention is 256, and the dimension of each embedding is 128. We employ a frozen *CLIP-ViT-L-14* model as our text encoder for text descriptions, and a learnable embedding for action categories. The shape of latent codes $z$ is set to 256, which is then injected into condition by concatenation for diffusion training and inference. Our models are trained with the AdamW optimizer using a fixed learning rate of $10^{-4}$. Our batch size is set to 4,096 during the INR training stage and 64 during the diffusion training stage separately. Since INR requires learning a latent code for each training sample, we set its batch size large for efficient training. Besides, INR model was trained for 20,000 epochs and diffusion model was trained for 3,000 epochs. The number of diffusion steps is 1,000 during training while 50 during inference. The corresponding variances $\beta_k$ in diffusion are scaled linearly from $8.5 \times 10^{-4}$ to 0.012. We train our models under Pytorch on NVIDIA GeForce RTX 3090.

## B.2    AUTO-DECODING OF LATENT CODE

Unlike traditional auto-encoder whose latent code is produced by the encoder, we draw inspiration from DeepSDF Park et al. (2019) in 3D shapes and use an auto-decoder to learn the latent code

Figure 6: Detailed network architecture of Codebook-Coordinate Attention (CCA).

without an encoder. This encoder-less design avoids explicit modeling of raw motions, making arbitrary-framerate training feasible. To construct distributions where one can sample a representation to generate a new high-quality motion, we gain insight from Variational Auto-Encoder (VAE) that utilizes an encoder to infer a distribution from which representations of motions can be sampled, and employs a decoder to reconstruct the data from the representation. As our design does not contain an encoder, we model each motion by sampling from a learnable distribution with optimized parameters, i.e., mean $\mu$ and covariance $\Sigma$, leading to learnable parameters $\{\mu_i, \Sigma_i\}_{i=1}^n$ for all training dataset. When training, we sample a latent representation $z_i$ from a distribution $\mathcal{N}(\mu_i, \Sigma_i)$. We then combine $z_i$ with temporal coordinate encodings $\gamma(t)$, and input them into a shared MLP decoder $f_\theta$. The parameters $\mu_i$, $\Sigma_i$ and $\theta$ are optimized simultaneously by minimizing the reconstruction loss between the generated and ground truth.

### B.3 SYNTHESIZING LONG SEQUENCES

Ideally, NeRM is possible to generate motions with arbitrary framerates $s$ and durations $l$ by setting appropriate temporal coordinates. However, either significant increase of $s$ or $l$ requires significant memory resources. Thus, we employ an iterative-synthesis approach to generate non-overlapping motion clips and assemble them into longer motions. By leveraging clip-based multi-framerate training conditioned on temporal coordinates, we learn continuous motion fields and ensure smooth transitions between motion segments. Notably, the framerate range we support is still limited by the original training data. The model is unlikely to learn motion patterns that exceed the highest framerate present in the training data.

## C  DETAILS OF EVALUATION METRICS

### C.1  EVALUATION METRICS

**Frechet Inception Distance (FID).** FID is widely used for overall generative quality evaluation. FID is calculated by extracting features from 1,000 generated motions and real motions in test set. Instead of the inception neural network in image domain, we extract a deep representation of the motion with the evaluator network Guo et al. (2022), as suggested by Tevet et al. (2023).

**R-Precision (Top-3).** This metric is used for text-motion matching measurement. For each generation, its corresponding GT description and randomly selected 31 mismatched descriptions are gathered in a pool. Then the Euclidean distance between the motion feature and each text feature in the pool is calculated, where accuracy of Top 3 are picked. The GT description falling into the Top 3 candidates represents a successful retrieval.

**MultiModality Dist.** Like R-Precision, this metric also measures text-motion similarity. We calculate the average Euclidean distance between the motion feature of each generation and text feature of the corresponding description.

**Diversity.** Diversity measures generative variance across all motions. We randomly select motions from all generations, and put each of them in either of the two subsets $\{q_1, \ldots, q_{D_a}\}$ and $\{q'_1, \ldots, q'_{D_a}\}$, with the same size as $D_a = 300$, where $q_i$ indicates a motion feature vector. Diversity can be expressed as:

$$\text{Diversity} = \frac{1}{D_a} \sum_{i=1}^{D_a} \|q_i - q'_i\|_2. \tag{7}$$

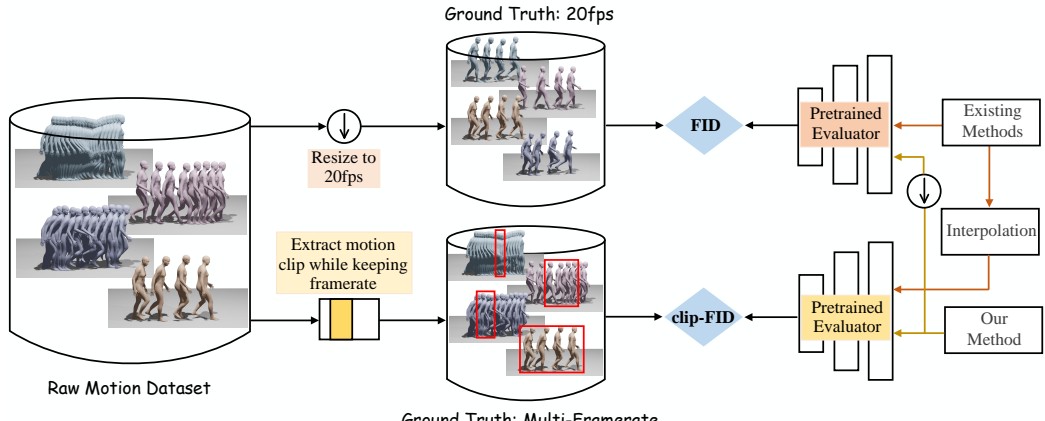

Figure 7: Comparisons of conventional FID and our clip-FID. FID evaluates global structure, but downsamples all human motions to a common 20fps which ignores high-framerate details. In contrast, clip-FID takes motion clips instead, thereby keeping the original framerates. We employ both metrics to validate the effectiveness of our method.

Table 4: Mean reconstruction errors of MLD and NeRM for motion of different framerates.

| Method / Metric | Framerates (fps) | | | | |
| --- | --- | --- | --- | --- | --- |
| | 20 | 40 | 60 | 100 | 120 |
| MLD / MPJPE | 0.027 | 0.062 | 0.113 | 0.184 | 0.228 |
| NeRM / MPJPE | **0.016** | **0.019** | **0.013** | **0.019** | **0.011** |
| MLD / MRE | 0.074 | 0.105 | 0.194 | 0.254 | 0.387 |
| NeRM / MRE | **0.041** | **0.036** | **0.034** | **0.035** | **0.038** |

**MultiModality (MM).** MM measures the generated motions diversify within each condition (text or action). We randomly pick $S$ text descriptions from all descriptions. Then the motions generated by the $s$-th description are randomly picked and put into one of the two subsets: $\{q_{s,1}, \ldots, q_{s,D_e}\}$ or $\{q'_{s,1}, \ldots, q'_{s,D_e}\}$, with subset size $D_e = 10$. MultiModality can be expressed as:

$$\text{MM} = \frac{1}{S \times D_e} \sum_{s=1}^{S} \sum_{i=1}^{D_e} \|q_{s,i} - q'_{s,i}\|_2. \tag{8}$$

**Accuracy.** We employ the pre-trained action recognition model to classify 1,000 generate motions. The obtained overall recognition accuracy demonstrates the correlation of the motion and its action label.

## C.2 MOTION CLIP-BASED FID (CLIP-FID).

All the metrics above are designed for low-framerate data, and unable to be trained on original high-framerate data due to prohibitive memory requirements. Thus, we here present a new metric clip-FID that is aimed at better capturing the realism of details at high framerate by avoiding downsampling. As depicted in Figure 7, we show the comparison of standard FID and clip-FID, where FID evaluates global structure **(the coarse information of the entire motion is preserved)**, but downsamples all human motions to a common 20fps, ignoring high-framerate details; clip-FID takes motion clips extracted/cropped from motions **(the detailed information of the motion clip is preserved)**, keeping the original framerates. Note that we randomly sample clips (a short segment of motion) of size $m$ from real motions at framerate $s$ and center $v$ to generate the corresponding clip. A lower value implies better high-framerate details.

Table 5: Ablation study on effectiveness of time encoding.

| Simple | Codebook | Motion Reconstruction (MRE) | | | | | Motion Synthesis (clip-FID) | | | | |
|---|---|---|---|---|---|---|---|---|---|---|---|
| | | 20 | 40 | 60 | 100 | 120 | 20 | 40 | 60 | 100 | 120 |
| ✗ | ✗ | 0.134 | 0.141 | 0.164 | 0.094 | 0.091 | 0.471 | 0.803 | 1.070 | 1.769 | 2.944 |
| ✓ | ✗ | 0.053 | 0.049 | 0.039 | 0.057 | 0.043 | 0.397 | 0.519 | 0.701 | 0.142 | 1.717 |
| ✗ | ✓ | **0.041** | **0.036** | **0.034** | **0.035** | **0.038** | **0.389** | **0.493** | **0.680** | **0.903** | **1.315** |

Table 6: Ablation study on effectiveness of Variational INRs.

| Variational INRs | Motion Reconstruction (MRE) | | | | | Motion Synthesis (clip-FID) | | | | |
|---|---|---|---|---|---|---|---|---|---|---|
| | 20 | 40 | 60 | 100 | 120 | 20 | 40 | 60 | 100 | 120 |
| ✗ | **0.032** | **0.030** | **0.031** | **0.036** | **0.027** | 1.280 | 2.924 | 7.012 | 10.482 | 14.654 |
| ✓ | 0.041 | 0.036 | 0.034 | 0.035 | 0.038 | **0.389** | **0.493** | **0.680** | **0.903** | **1.315** |

# D  NETWORK ARCHITECTURE

**Codebook-Coordinate Attention (CCA).** Inspired by CoCo-NeRF (Yin et al., 2022), we apply CCA modulation to enrich the Fourier features of each coordinate. The detailed architecture is illustrated in Figure 6. Specifically, we employ one cross-attention block to learn dependency between learnable query embeddings $\mathcal{Q} = \{q_i\}_{i=1}^{M}$ and codebook prototypes $\mathcal{E} = \{e_i\}_{i=1}^{N}$. Then, the embeddings are fed into self-attention blocks with three layers to improve their feature representations further and obtain the final motion-relevant prototypes $\hat{\mathcal{Q}}$. Finally, we conduct once cross-attention operation between $\hat{\mathcal{Q}}$ and Fourier embeddings of each coordinate $\gamma(t)$. All attention modules are based on transformer (Vaswani et al., 2017) with 4 head Attention mechanism, Layer Normalization, Feed-Forward Network and GELU activation.

**MLP Decoder.** Similar to other neural representations (Ashkenazi et al., 2023), the NeRM decoder $f_\theta$ is constructed using a simple neural network architecture. It consists of a 9-layer MLP with ReLU activations and layer normalization. The hidden layer size remains constant across the network. We also incorporate residual connections within each layer to improve gradient flow. In contrast to previous approaches (Chen et al., 2023; Tevet et al., 2023) that rely on a transformer backbone, our choice of a simple decoder offers benefits in terms of inference speed for generating new motions.

**Latent Denoiser.** Different from the UNet-based architecture (Ronneberger et al., 2015) in latent diffusion model that designed for image synthesis, our latent denoiser $\epsilon_\phi$ is built ViT backbone with long skip connections (Bao et al., 2023), which is more appropriate for time series data.

# E  ADDITIONAL EXPERIMENTS

Our NeRM consists of an INR model $f_\theta$ and a latent diffusion model $\epsilon_\phi$. We conduct additional experiments to evaluate the effectiveness of each component, as well as reconstruction capability. We also report time costs on inference and GPU memory on multi-framerate training dataset.

## E.1  RECONSTRUCTION CAPABILITY

We first compare the reconstruction capability of MLD (Chen et al., 2023) and our NeRM with different framerates. Note that MLD learns latent representation via Variational AutoEncoder (VAE) based on transformer encoder. We randomly select 100 different motion sequences with their original framerate on HumanML3D dataset, and report the average reconstruction errors in Table 4. Since MLD cannot process multi-framerate dataset, we use spherical linear interpolation to generate higher-framerate motions for MLD. For evaluation metrics, we make use of the Mean Per Joint Position Error (MPJPE) (Mao et al., 2019), commonly used for image-based 3D human pose estimation. We also employ Mean Redundant Error (MRE) that measures the Euclidean distance between two

Table 7: Ablation study on decoder architecture.

| MLP | Transformer | Motion Reconstruction (MRE) | | | | | Motion Synthesis (clip-FID) | | | | |
|---|---|---|---|---|---|---|---|---|---|---|---|
| | | 20 | 40 | 60 | 100 | 120 | 20 | 40 | 60 | 100 | 120 |
| ✗ | ✓ | **0.037** | **0.032** | 0.037 | **0.035** | **0.037** | **0.381** | **0.487** | **0.607** | **0.782** | **1.199** |
| ✓ | ✗ | 0.041 | 0.036 | **0.034** | **0.035** | 0.038 | 0.389 | 0.493 | 0.680 | 0.903 | 1.315 |

Table 8: Ablation study on the dimension $d$ and weight parameter $\lambda_{KL}$ of latent representation.

| $d$ | 128 | 256 | 512 |
|---|---|---|---|
| **MRE@20fps** | 0.0459 | 0.0412 | **0.0407** |
| $\lambda_{KL}$ | 1e-5 | 1e-4 | 1e-3 |
| **MRE@20fps** | **0.0410** | 0.0412 | 0.0459 |

poses represented by redundant hand-crafted features. From the results in Table 4, we find that our NeRM with a simple MLP can achieve lower reconstruction error with 20 fps than MLD with complicated transformer backbone. When the framerate increases, the performance of MLD deteriorates significantly while NeRM still maintains very low reconstruction error.

### E.2 ABLATION STUDY

In this section, we provide ablation studies on the HumanML3D dataset to evaluate the effectiveness of network design.

**Effectiveness of Time Encoding.** Time encoding plays a critical role in the generalization. In Table 5, we evaluate the influence of different encoding manners, including No Time Encoding, Simple Time Encoding, and Codebook-Coordinate Encoding. Here, "Simple" means that we only use Fourier features to encode $t$ like traditional NeRF (Mildenhall et al., 2021); and "Codebook" is our design of NeRM. From the table, we find that the model without time encoding achieves poor performance in both terms of motion reconstruction and motion synthesis. This can be attributed to "spectral bias" (Rahaman et al., 2019). In other words, INRs with simple MLP layers cannot learn high-frequency variations from motion data. Our codebook-based representation reaches the best performance, which confirms that the codebook is beneficial for feature representation of temporal coordinates.

**Effectiveness of Variational INRs.** We investigate the influence of variational INR by comparing a variational INR with a non-variational version. In this case, the latent code $\boldsymbol{z}$ of non-variational INR is obtained by optimizing the following:

$$\boldsymbol{z}_i^* = \arg\min_{\boldsymbol{z}_i} \|\hat{x}_{clip}^i - x_{clip}^i\|^2, \quad \text{for } i = 1, 2, \cdots, n \tag{9}$$

where $\hat{x}_{clip}^i = f_\theta(t_{\boldsymbol{v},s}, s)$. Comparison results are shown in Table 6. We observe that the reconstruction errors of non-variational INRs is smaller than variational ones, as non-variational INRs may overfit the motion sequence by powerful neural network. However, the realism of non-variational INRs is significantly improved by the variational ones. This suggests that the variational approach strongly regularizes the latent space and enhances the capability of sampling new motions.

**Decoder Architecture.** Furthermore, we compare the performance of MLP-based and transformer-based models. Table 7 shows that the self-attention mechanism can slightly improve the ability of motion synthesis. However, the latent representation learned by diffusion model is fed to the decoder of our variational INR, which implies that the complexity of the network greatly impacts the generation speed of new motions. Therefore, we pick MLP layers as our decoder architecture.

**Effectiveness of latent representation.** The latent representation $\boldsymbol{z}$ is a crucial variable in our NeRM. It acts as a bridge between the variational INR model $f_\theta$ and the diffusion model $\epsilon_\phi$. In our context, the quality of latent representation can be influenced by the dimension and the variational distribution $\mathcal{N}(\mu, \Sigma)$ of $\boldsymbol{z}$. Therefore, we investigate the impact of different dimensions $d$ of $\boldsymbol{z}$ and weight $\lambda_{KL}$ on the KL loss for motion reconstruction in Table 8. When the dimension $d$ is set to

Table 9: Ablation study on effectiveness of time normalization.

| Time Normalization | MRE@20fps | FID | Diversity |
|:---:|:---:|:---:|:---:|
| ✗ | 0.118 | 0.958 | 9.892 |
| ✓ | **0.041** | **0.389** | **9.547** |

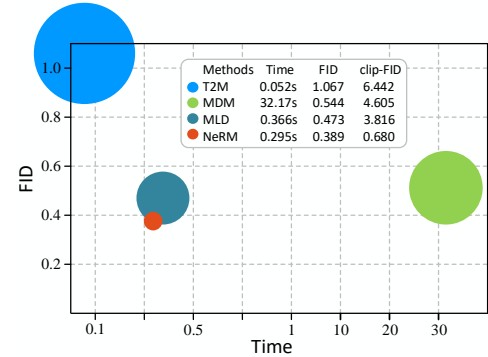

Figure 8: Average inference time (seconds) for generating one motion sequence. The circle size is proportional to the value of clip-FID. Bigger circle indicates worse performance of high-framerate details.

256, the motion reconstruction loss becomes significantly small, and higher dimension can slightly improve the reconstruction ability. For $\lambda_{KL}$, a higher weight results in a smoother latent space but increases the reconstruction error. We set $\lambda_{KL}$ to a small value (i.e., 0.0001) like Chen et al. (2023).

**Effectiveness of time normalization.** In this section, we explore the effect of time normalization. Table 9 shows the results on HumanML3D dataset with 20 fps. Notably, '✗' indicates that we directly use the true temporal coordinates, i.e., $\{1, 2, \cdots, T\}$, instead of normalized temporal coordinates $t_{v,s}$. From the table, we find that time normalization plays a vital role in time encoding of implicit neural representation.

### E.3 INFERENCE TIME

In Figure 8, we report average inference time on per sequence. As T2M (Guo et al., 2022) is built under VAEs, it uses the least time for generation; but under diffusion setting, we are the most time-saving. To be specific, due to the latent diffusion design, MLD (Chen et al., 2023) and our NeRM are much faster than MDM (Tevet et al., 2023). We are even faster than MLD in that we use simple MLP decoder of INR, rather than the transformer-based decoder in MLD. We yield the best FID for best global motion quality, and significant superiority in clip-FID (the target framerate is set to 60 fps), indicated by the smallest circle size. All of these experiments are conducted on NVIDIA GerForce RTX 3090.

### E.4 MEMORY BURDEN

By padding zeros, current text-to-motion generative models are able to train motions with the same framerate $s$ (20 fps) and different duration $2 \leq l \leq 9.8$ (seconds). When exploiting native framerates of motions, one possible solution is to padding zeros according to the maximum framerates (250 fps). However, this operation cannot capture accurate temporal dependency as fixed-framerate training. Another problem is that the dimensionality of padded motions needs to be much larger than what state-of-the-art diffusion models can be trained on. For example, MDM (Tevet et al., 2023) trains the diffusion model on motions where the maximum number of poses is 196 ($9.8s \times 20fps$), while we use motions of maximum size 2450 ($9.8s \times 250fps$). This imposes a substantial burden on GPU memory. Alternatively, we approach it from the perspective of implicit neural representation, which enables us to process *clips* (a short segment of motion) with size $m$ ($m \ll 2450$), therein significantly reducing the memory burden. In addition, we find a representative latent code for each

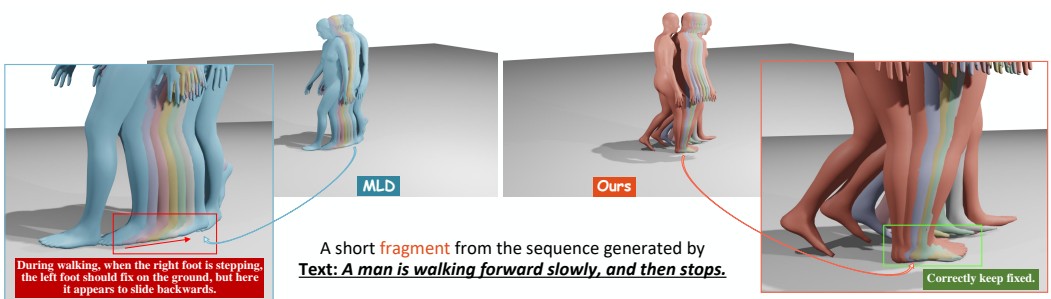

Figure 9: High-framerate generation (120 fps). We visualize a short fragment from the whole generation, with purple, pink, yellow, green, and grey sequentially indicating high-framerate motion changing. The entire generation can be found in our video.

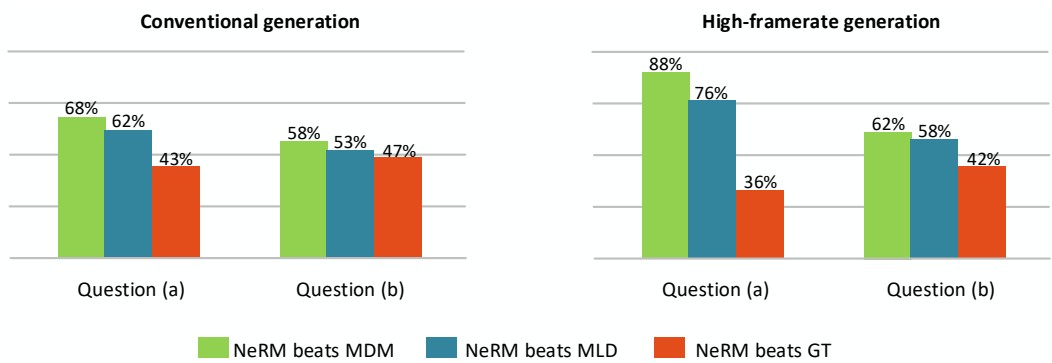

Figure 10: User study on HumanML3D dataset.

motion and learn the distribution in this low-dimensional latent space efficiently, which decouples the modeling of the distribution from varied-size human motions.

### E.5 VISUALIZATION

A video is contained in our supplement. We provide 120 fps generation on two more examples, where our NeRM constantly outperforms MLD (Chen et al., 2023) in both terms of basic motion quality and high-framerate quality. Note that, as MLD cannot directly generate high-framerate motions, we use SLERP to interpolate them towards the target framerate (120 fps). We also select a short fragment from our video with full discussion shown in Figure 9. Additionally, for clearer observation on artifacts of baseline, we slow down the video by 6 times, so the motions may appear to be slower.

### E.6 USER STUDY

Human eyes are the ultimate evaluation for human motion performance. We asked 17 people over question (a) "Which of the two motions is more realistic" and (b) "Which of the two motions is more consistent with the given texts". In each *two-motion* pair, we provide one motion generated by NeRM and the other by baselines MDM (Tevet et al., 2023), MLD (Chen et al., 2023) and GT from HumanML3D. We randomly pick 8 cases for each question. We evaluate on conventional 20 fps generation and high-framerate generation (each of 120, 100, 60 and 40 fps has 2 cases). Shown in Figure 10, our NeRM gains more preference than baselines, and even comparable to GT motions.

## F    LIMITATIONS AND FUTURE WORK

Although our method has achieved promising performance for high-framerate motion synthesis, it still has the following challenges. (1) While our method currently supports generation with external conditions such as action labels or text prompts, it has limitations in incorporating fine-grained internal conditions such as keyframes or trajectory. A promising direction for future research is to design a more comprehensive framework (as exemplified by Karunratanakul et al. (2023); Zhang et al. (2023b)) that can simultaneously consider both external and internal conditions. By integrating these factors, we can achieve more precise and nuanced motion generation. (2) The quality of motion generated using INRs is highly dependent on the dataset. If the dataset lacks high-framerate data, the performance of generating high-framerate motions may not be optimal. Additionally, the model is unlikely to learn motion patterns that exceed the highest framerate present in the training data. (3) While our method demonstrates fast inference time, the training process can be relatively slow, particularly when dealing with the dataset containing numerous samples. This is because our method learns a latent code for each training sample. (4) Our method is designed for motion modeling and can not adapt to outputs with varying body shapes. Some recent works Mihajlovic et al. (2022) have employed INRs to model skinned articulated objects with specific poses. It would be interesting to explore the integration of our method with them.

