# OpenReview forum: "NeRM: Learning Neural Representations for High-Framerate Human Motion Synthesis"
_ICLR.cc/2024/Conference — ICLR 2024 poster_

### Official Review · Reviewer_WpJV · 2023-10-28

**Soundness:** 3 good
**Presentation:** 3 good
**Contribution:** 3 good
**Rating:** 8
**Confidence:** 4

**Summary:**

The paper introduces NeRM, a generative model for high-framerate human motion synthesis using Implicit Neural Representations (INRs).
NeRM can handle varied-size data and capture the variational distribution of motions for high-framerate motion synthesis.

**Strengths:**

The paper provides a clear and concise description of the problem statement, methodology, and evaluation metrics. The paper addresses the underexplored task of generating realistic human motions with high framerates. By leveraging the advantages of INRs and diffusion models, NeRM offers a memory-friendly and efficient solution for high-framerate motion synthesis.

**Weaknesses:**

* The paper could provide more detailed explanations and insights into the limitations and challenges of using Implicit Neural Representations (INRs) for high-framerate motion synthesis. This would help readers understand the potential trade-offs and constraints associated with the proposed approach.

* The paper could benefit from a more extensive discussion on the generalizability of NeRM to different datasets and motion types. It would be valuable to explore the performance of NeRM on diverse motion datasets and evaluate its ability to handle complex and varied motion patterns

* Some more motion synthesis literatures can be included in this paper, such as:
[a] A unified 3d human motion synthesis model via conditional variational auto-encoder
[b] Towards diverse and natural scene-aware 3d human motion synthesis.

**Questions:**

It would be beneficial to include more detailed explanations and insights into the proposed clip-FID metric for evaluating the quality of high-framerate generative details. How does clip-FID preserve target framerates without downsampling and how does it capture local details and artifacts such as foot sliding?

---

> ### Author Response · Authors · 2023-11-18
> **Response to Reviewer WpJV (part 1)**
>
> We thank the reviewer for the positive comments. We are glad that the reviewer finds our motivation strong and the method clear. Below we address all the concerns.
>
> _**W1: The paper could provide more detailed explanations and insights into the limitations and challenges of using INRs for high-framerate motion synthesis. This would help readers...**_
>
> We thank the reviewer for this great suggestion and summarize the limitations and challenges here.
>
>  (1) Although our method currently supports generation with external conditions such as action labels or text prompts, it has limitations in incorporating fine-grained internal conditions such as keyframes or trajectory. A promising direction for future research is to design a more comprehensive framework (as exemplified by [Ref1, Ref2]) that can simultaneously consider both external and internal conditions. By integrating these factors, we can achieve more precise and nuanced motion generation.
>
> (2) The quality of motion generated using INRs is highly dependent on the dataset. If the dataset lacks high-framerate data, the performance of generating high-framerate motions may not be optimal. Additionally, the model is unlikely to learn motion patterns that exceed the highest framerate present in the training data.
>
> (3) While our method demonstrates fast inference time, the training process can be relatively slow, particularly when dealing with the dataset containing numerous samples. This is because our method learns a latent code for each training sample.
>
> (4) Our method is designed for motion modeling and can not adapt to outputs with varying body shapes. Some recent works [Ref3] have employed INRs to model skinned articulated objects with specific poses. It would be interesting to explore the integration of our method with them. These limitations and future work can be found in Sec. F of Appendix.
>
> _**W2: The paper could benefit from a more extensive discussion on the generalizability of NeRM to different datasets and motion types. It would be valuable to explore...**_
>
> We thank the reviewer for the thoughtful comment. We will add a quadruped motion dataset [Ref4] that contains 30 minutes of dog motions. Just a kindly note that, we have covered almost all the mainstream datasets that applied in current human motion generation (text-to-motion, action-to-motion, unconditional generation) tasks, and achieved promising results on them, which can validate the effectiveness of our approach. However, we really appreciate the reviewer’s enlightment of including more interesting cross-domain datasets. We are now experimenting on [Ref4], which takes much time since it refers to a different skeleton representation and we need to conduct further research over pre-processing and training stuff. We will include the results in the final version.

---

> ### Author Response · Authors · 2023-11-18
> **Response to Reviewer WpJV (part 2)**
>
> _**W3: Some more motion synthesis literatures can be included in this paper, such as: [a] A unified 3D human motion synthesis model via conditional variational auto-encoder [b]...**_
>
> We thank the reviewer for the suggestion of the two great works [Ref5, Ref6] as references. We carefully include them into the Introduction (Sec. 1 of main paper) with detailed descriptions in Sec. A of Appendix.
>
> _**Q1: It would be beneficial to include more detailed explanations and insights into the proposed clip-FID metric for evaluating the quality of high-framerate generative details. How does...**_
>
> We thank the reviewer for the suggestion. Following the reviewer's comment, we add a figure (Figure 2 in Sec. C.2 of Appendix) and related description (in Sec. C.2 of Appendix) to better explain the proposed clip-FID metric. Specifically, the figure exhibits the comparison of standard FID and clip-FID, where FID evaluates global structure **(the coarse information of the entire motion is preserved)**, but downsamples all human motions to a common 20fps, ignoring high-framerate details; clip-FID takes motion clips extracted/cropped from motions **(the detailed information of the motion segment is preserved)**, keeping the realism of details at high framerate without downsampling. Note that we randomly sample clips (a short segment of motion) of size $m$ from real motions at framerate $s$ and center $v$ to generate the corresponding clip.
>
> Due to downsampling, the pre-processed low-framerate data in standard FID inevitably misses the expressive details in motions (e.g., how does the foot move when lifting it during walking) or extreme poses (e.g., in the process of extending and retracting the arm, the farthest reaches of hand may be discarded due to down-sampling, especially if the movement is fast). On the other hand, as each frame of the original motion is not discarded in clip-FID, local details and artifacts such as foot sliding are captured.
>
> **References:**
>
> [Ref1] Karunratanakul et al. Guided Motion Diffusion for Controllable Human Motion Synthesis. ICCV, 2023.
>
> [Ref2] Zhang et al. FineMoGen: Fine-Grained Spatio-Temporal Motion Generation and Editing. NeurIPS, 2023.
>
> [Ref3] Mihajlovic  et al. COAP: Compositional Articulated Occupancy of People.  CVPR, 2022.
>
> [Ref4] Zhang et al. Mode-adaptive Neural Networks for Quadruped Motion Control.  ACM Trans. Graph, 2018.
>
> [Ref5] Cai et al. A Unified 3D Human Motion Synthesis Model via Conditional Variational Auto-Encoder. ICCV, 2021.
>
> [Ref6] Wang et al. Towards Diverse and Natural Scene-Aware 3D Human Motion Synthesis. CVPR, 2022.

---

### Official Review · Reviewer_KnYZ · 2023-10-29

**Soundness:** 2 fair
**Presentation:** 3 good
**Contribution:** 2 fair
**Rating:** 5
**Confidence:** 4

**Summary:**

This paper focuses on the human motion synthesize for frame rate scenarios. It recognizes the limitations of previous efforts of generating high-frame-rate human motion sequences. The key idea is to fuse the data of different framerates into training by normalizing the time positions into relative and centerized time indices with a continuous mapping from time position to the pose configurations. A progressive training is leveraged to relieve the pressure of learning motion patterns under different frame rates by bootstrapping from the fixed frame rate. The generation of motion sequences is done by a conditional diffusion model where the latent code is the motion code from encoders and a codebook-based attention module.

**Strengths:**

- The proposed method can accept training data in arbitrary frame rate while supporting human motion generation in high frame rate.
- The proposed method is flexible to support human motion generation of different schemes, such as unconditional, or conditional to different constraints, such as action label or text descriptions.
- Leveraging the latent code for diffusion model and integrating the codebook-based attention module, the method is designed in a whole to support different modalities for conditions and good expressiveness.

**Weaknesses:**

I put both my recognized weaknesses and the questions I would like authors’ response to here.

- Question: Eq 2 is confusing to me. Normalizing the clip length from the standard seconds to the relative length has been commonly adopted. In Eq 2, given both the relative time position and the frame rate, the input information to the generation function f_\theta is exactly the same as what previous works input. Could the authors elaborate more about their differences here?
- Question: with different frame rates for the target motion sequence, is the number of time steps in diffusion, i.e. k in Eq 5, the same? I understand that they are two different “time steps” but I still would like to get a sense that whether the diffusion model is able to capture the motion of different frame rates or it is only to recover the pose in a single time step, static.
- In my understanding the claim that “NeRM can generate motions with arbitrary framerates s and durations l by setting appropriate temporal coordinates.” may be inaccurate. Technically, by the design of the method, the claim can be partially correct. But given the training data captured under certain frame rates, without considering the motion speed and sensitiveness of capture sensor etc, the model is unlikely to learn motion patterns in frame rates exceeding the highest frame rate contained in the training data.
- Question: is there any reason that a close baseline Nemf (He et al 2022) is not included in the benchmarking comparisons?
- The method is constructed with multiple components and requires settings of many introduced hyper-parameters, such as the normalizing of time positions, the codebook-coordinate attention and etc. The authors may need to provide corresponding ablation studies to support the effectiveness of these modules and the help readers understand the resources of performance gains more clearly.

**Questions:**

Please see my questions above.

---

> ### Author Response · Authors · 2023-11-18
> **Response to Reviewer KnYZ (part 1)**
>
> We thank the reviewer for the insightful comments. We are glad that the reviewer finds our method flexible, expressive and support different modalities. Below we address all the concerns.
>
> _**Q1: Eq 2 is confusing to me. Normalizing the clip length from the standard seconds to the relative length has been commonly adopted...**_
>
> We thank the reviewer for this great reminder, and elaborate more about the differences between previous works (including motion-based INRs and image-based INRs) input and ours in Eq 2. We revise the related content in Sec. 3.1 in main paper. First, we sincerely apologize that the input to Eq 2 needs to be revised as $(t_{v,s}^{i},z_{i},s^{i})$. Below we provide detailed explanation.
>
> **(1) The input to motion-based INRs vs. ours.** Previous motion-based INRs only consider the simple setting: all $n$ training motions $\textbf{X}=[x^{1},x^{2},\cdots,x^{n}]$ in a dataset have the same framerate $s$ and duration $l$, leading to the fixed number of human poses (i.e., $T=s\cdot l$) in each motion. In this case, they regard a motion $ x^{i}=[x_{1}^{i},x_{2}^{i},\cdots,x_{T}^{i}]\in\mathbb{R}^{d\times T}$ as a function of temporal coordinates $t=[t_{1},t_{2},\cdots,t_{T}]$, i.e., $f_{\theta}: (t_{j},z_{i})\mapsto x_{j}^{i}$ for any $j=1,2,\cdots,T$ and $i=1,2,\cdots,n$. To normalize the coordinates into a bounded domain $[-1,1]$, their normalized temporal coordinate can be represented by $t=[-1,\frac{-T+2}{T},\frac{-T+4}{T},\cdots,\frac{T-2}{T},1]$. **The interval of each two normalized coordinate is** $\frac{2}{T}$. **Note that all motions in the dataset have the same normalized temporal coordinate** $t$. However, in our setting, all training motions $\textbf{X}=[x^{1},x^{2},\cdots,x^{n}]$ in a dataset have different framerate $s^{i}$ and duration $l^{i}$, resulting in varied number of human poses (i.e., $T^{i}=s^{i}\cdot l^{i}$). This task is quite challenging as the network architecture can only accept mini-batch data of a fixed size. Thus, we randomly choose a motion clip of a fixed size $m$ for each motion, obtaining $[x_{clip}^{1},x_{clip}^{2},\cdots,x_{clip}^{n}]$ with each motion clip $x_{clip}^{i}\in\mathbb{R}^{d\times m}$. Now, we have a fixed-size motion clips like existing motion-based INRs, but we cannot adopt similar coordinate normalization because each motion clip comes from different position of the original motion. Therefore, we propose the following method. Suppose that the maximum duration is $l_{max}$ for all motions,  we normalize the temporal coordinates of the $i$-th motion as $t_{s}^{i}=[-1,\frac{-s^{i}\cdot l_{max}+2}{s^{i}\cdot l_{max}}, \frac{-s^{i}\cdot l_{max}+4}{s^{i}\cdot l_{max}},\cdots,\frac{s^{i}\cdot l_{max}-2}{s^{i}\cdot l_{max}},1]$. **The interval of each two normalized coordinate is** $\frac{2}{s^{i}\cdot l_{max}}$. **Note that all motions in the dataset have the different normalized temporal coordinate** $t$, **depending on the framerate** $s^{i}$. Our extracted motion clip also depends on the center $v$. Finally, if we want to extract the first $m$ poses as a motion clip from the $i$-th motion with framerate $s^{i}$ and duration $l^{i}$, the normalized coordinates can be represented as $t_{v,s}^{i}=[-1,\frac{-s^{i}\cdot l_{max}+2}{s^{i}\cdot l_{max}}, \frac{-s^{i}\cdot l_{max}+4}{s^{i}\cdot l_{max}},\cdots,\frac{-s^{i}\cdot l_{max}+2\cdot (m-1)}{s^{i}\cdot l_{max}}]$. Due to different $s^{i}$ in each motion, we also input it as a condition. The input of Eq 2 becomes $(t_{v,s}^{i},z_{i},s^{i})$, which is totally different from previous works.
>
> **(2) The input to image-based INRs vs. ours.** Previous image-based INRs regard an image as a function of horizontal and vertical coordinate $(p_{x},p_{y})$ and maps to RGB values. The images in the dataset have the fixed resolution $r\times r$. Therefore, their normalized spatial coordinate can be represented by $p_{x}=[-1,\frac{-r+2}{r},\frac{-r+4}{r},\cdots,\frac{r-2}{r},1]$ and $p_{y}=[-1,\frac{-r+2}{r},\frac{-r+4}{r},\cdots,\frac{r-2}{r},1]$, leading to a bounded domain $[-1,1]\times [-1,1]$. **The interval of each two normalized coordinate is** $\frac{2}{r}$. If each image in the dataset has its own resolution $r^{i}$, the interval can be formulated as $\frac{2}{r^{i}}$. Images are continuous in 2D space, and motion sequences are continuous in the temporal dimension, but we can not simply handle this difference by ignoring one dimension. The **essential difference between image and motion** lies in that, image size (pixel number) is determined only by resolution, while motion size (frame number) is determined by both framerate and duration. This leads to the different normalized coordinate setting: $\frac{2}{r^{i}}$ for image while $\frac{2}{s^{i}\cdot l_{max}}$ for motion. (Note that resolution in image can be analogous to framerate in motion.)

---

> > ### Comment · Reviewer_KnYZ · 2023-11-22
> > **Feedback (part 1)**
> >
> > Thanks for the reply.
> >
> > Yes, your revision of the draft makes the formulation understandable now.
> >
> > So basically, in the normalization of temporal dimensions, you keep the number of time strides the same while the length of each stride is conditional to an extra condition parameter $s$. This makes sense to me.
> >
> > However, when considering the novelty of this property, I noticed a claim in your paper:
> >
> > >  We find these latent
> > codes by interpreting them as a variational distribution with optimized parameters (i.e., mean and
> > covariance) in the representation space. This simple process naturally supports any-framerate training and showcases a significant departure from previous works (He et al., 2022; Mao et al., 2021)
> > that require an explicit and sophisticated encoder to infer distributions of fixed-framerate motions.
> >
> >
> > However, if my understanding is correct, NeMF (He et al.) also supports the motion inference under arbitrary frame rate. Quoted from its paper:
> >
> > > Unlike other motion models, NeMF is theoretically guaranteed to generate smooth motion
> > in arbitrary frame rates.
> >
> > Could you further elaborate on the novelty of your method's support of non-fixed framerate motion generation over NeMF?

---

> > > ### Author Response · Authors · 2023-11-22
> > > **Reply to Feedback (part 1)**
> > >
> > > Thank you for your kind reply.
> > >
> > > NeMF (He et al., 2022) indeed supports arbitrary-framerate, **but only at _inference_**, which can be attributed to the inherent property of INR. However, it requires an explicit and sophisticated encoder $\phi$ of VAE to infer the distribution of the latent variable $z_{i}$. This encoder $\phi$ only accepts the fixed-size of mini-batch data (e.g., 16[batchsize]$\times$196[motion length]$\times$263[feature]). Consequently, **it becomes infeasible to utilize multi-framerate training data.** In this way, NeMF has to downsample training data to a fixed and lower framerate, like other conventional models. On the other hand, our design regards the latent variable as a variational distribution with optimized parameters, avoiding an explicit encoder. This means that we no longer need to downsample motions to form the fixed-size training data, and therefore **naturally support arbitrary-framerate _training_**. This is significantly beneficial as current annotated motion data are limited and costly. As a result, we can fully utilize the raw data without any information loss, to produce more realistic motions.
> > >
> > > If you have further questions, we are happy to discuss them.

---

> ### Author Response · Authors · 2023-11-18
> **Response to Reviewer KnYZ (part 2)**
>
> _**Q2: With different framerates for the target motion sequence, is the number of time steps in diffusion, i.e. $k$ in Eq 5, the same? I understand...**_
>
> Yes, the number of time steps in diffusion is the same for the target motion sequence with different framerates. For motion sequences with different framerates and durations, our first stage (variational INR) is dedicated to reconstructing these challenging motion sequences, thereby learning **a representative "global" latent variable $z$** for each motion. Notably, the global latent variable is concatenated to each time step during the training of INR. Consequently, the latent variable is naturally responsible for recovering all time steps in the decoder part. Our second stage (conditional diffusion) primarily focuses on learning the distribution of these global variables,  where each global variable corresponds to a complete motion sequence. Moreover, our variational INR approach helps smoothen the latent space of $z$, which facilitates the training of the diffusion model. This mechanism shares similarities with the framework of Stable Diffusion [Ref 1]. It should be emphasized that while the global latent vector can create a bottleneck for the network, it does not imply that latent vector can only recover a single static pose.
>
> Overall, it is the variational INR stage that is responsible for capturing the motion of different framerates and durations, as it learns to reconstruct the motions and recover poses in all time steps. The diffusion model, on the other hand, focuses on learning the distribution of the latent variables obtained from the variational INR stage. We revise the related description in Sec. 3 in the manuscript.
>
> _**Q3: In my understanding the claim that "NeRM can generate motions with arbitrary framerates and durations by setting appropriate temporal coordinates" may be inaccurate...**_
>
> We agree with the reviewer that the model is unlikely to learn motion patterns in framerates exceeding the highest framerate contained in the training data. Just a kindly reminder, actually, we have claimed such limitation in the last lines of Sec. 3.2 in the original paper (Due to space limitation, we move it to Sec. B.3 in Appendix). We carefully improve our expression to make it clearer according to the reviewer's suggestion and highlight it with blue. We also involve it as a limitation in Sec. F of Appendix.
>
> _**Q4: Is there any reason that a close baseline NeMF (He et al. 2022) is not included in the benchmarking comparisons.**_
>
> We appreciate the reviewer for the valuable question. Our NeRM is designed for motion generation with external conditions like text prompts and action labels, so the baselines we choose are also under the same setting. Differently, NeMF focuses on using internal conditions (like keyframes that drawn from the motion itself) for generation, which is orthogonal to our setting. Meanwhile, NeMF can only realize single-motion (deterministic) generation based on its conditions, and cannot support diverse (probabilistic) generation like ours.
>
> However, considering that both NeRM and NeMF have the compatibility of handling unconditional motion generation, we refer to the results (on AMASS dataset) in original NeMF paper, and compare it with ours as below. It illustrates that our NeRM outperforms NeMF in terms of both FID and diversity. This can be ascribed to flexible multi-framerate training and powerful generative ability of the diffusion model. Please note that NeMF is based on VAE framework and can only support fixed-framerate training. We include them in Sec. 4.3 of main paper.
>
> | Method | FID&darr;  | Diversity&uarr; |
> | ------ | ------------------------------------------------- | ------------------------------------------------------ |
> | NeMF   | 2.07                  | 2.77      |
> | MDM    | 8.84                  | 6.43     |
> | MLD    | 1.40                  | **8.57**      |
> | NeRM   | **1.07**                  | 6.91     |
>
> _**Q5: The method is constructed with multiple components and requires settings of many introduced hyper-parameters, such as the normalizing of time positions...**_
>
> We thank the reviewer for this great suggestion. Following your comment, we will add the ablation study of normalizing of time positions in Sec. E.3 of Appendix. Ablation studies on the time position encodings, codebook-coordinate attention and some other components have been involved in Table 2, Sec. E.3 of Appendix. Please let us know if you have further requirements for ablation of our components.
>
> **References:**
>
> [Ref1] Rombach et al. High-Resolution Image Synthesis with Latent Diffusion Models. CVPR, 2022.

---

> > ### Comment · Reviewer_KnYZ · 2023-11-22
> > **Feedback (Part 2)**
> >
> > Thanks for the reply!
> >
> > I carefully read your reply and believe these clarifications and the related revision are helpful for the audience, of course including me, to understand the contribution and limitations of this paper.

---

### Official Review · Reviewer_9SXt · 2023-10-30

**Soundness:** 3 good
**Presentation:** 3 good
**Contribution:** 3 good
**Rating:** 8
**Confidence:** 4

**Summary:**

This paper proposes a neural representation, i.e., NeRM, for representing continuous human motions. NeRM directly learns a continuous motion field over temporal coordinates without explicit modeling, making the training with varied-framerate motions and high-framerate motion generation possible. The authors leverage the proposed representation to (un-)conditional motion generation with diffusion models, showing the efficiency and effectiveness of high-quality motion generation.

**Strengths:**

The primary contribution of this work is the proposed neural continuous motion representation, NeRM, which I find both interesting and innovative. It effectively represents continuous motion sequences at any framerate level. This representation addresses limitations observed in previous works, such as MLD, which necessitates the input motion sequence to have the same framerate and fails to capture high-frequency details. The experiments conducted on various conditional and unconditional motion generation tasks are thorough and robust. The paper is well-organized and clearly presented.

**Weaknesses:**

My main concern revolves around the motivation of representing and directly generating high-framerate human motions. As discussed in the introduction, the authors present two key points: (1) high-framerate motion generation is inefficient, and (2) training with a fixed framerate cannot adequately utilize the dataset. However, I believe that training with fixed, low-framerate data might suffice to produce high-framerate results through interpolation. Hence, there might be no imperative need to use per-frame human poses during training.
The related discussion in the introduction appears to be not highly convincing.

**Questions:**

1. How can we determine if the feet sliding shown in Figure 4 is a result of training with a low framerate?

2. Does the use of variational INR, i.e., normalizing the latent code $\mathbf{z}_i$ to a normal distribution with KL loss, affect the preservation of high-framerate details? Such normalization typically leads to a smoother representation space?

---

> ### Author Response · Authors · 2023-11-18
> **Response to Reviewer 9SXt (part 1)**
>
> We thank the reviewer for the positive comments. We are glad that the reviewer finds our method interesting and innovative, the experiments thorough and robust. Below we address all the concerns.
>
> _**W1: Concerning about the necessity of representing and directly generating high-framerate human motions.**_
>
> We need to politely emphasize that, the ability to represent and directly generate high-framerate motions is worth exploring, and training on high-framerate data is also necessary.
>
> **In terms of high-framerate representing.** Let us see Microsoft Surface Pro 9 2-in-1 tablet (120Hz refresh rate), Asus ROG 27-inch 2K native esports display (360Hz), Apple Vision Pro (90/96/100Hz) and many other devices with high refresh rate. The popularity of these devices among electronics enthusiasts indeed reflects the huge demand for high-framerate representation/display.
>
> **In terms of direct high-framerate generation.** This technology is useful, which cannot be replaced by low-framerate generation attached with interpolation due to the following reasons:
> The pre-processed low-framerate data inevitably misses the expressive details in motions (e.g., how does the foot move when lifting it during walking) or extreme poses (e.g., in the process of extending and retracting the arm, the farthest reaches of hand may be discarded due to down-sampling, especially if the movement is fast). If generative models are trained with such kind of low-framerate motion data, then their low-framerate generations certainly cannot re-create the ideal intermediate content by interpolation, as interpolation can only produce monotonous motions and often causes unrealistic sliding phenomenon.
> Moreover, our NeRM enables more than just high-framerate generation, but varied-framerate generation thanks to INR design. It means that we are compatible to devices with any refresh rate. If it is high, we are able to; if it is low, we can also adapt to it. However, for existing low-framerate generators, when faced with changes of target framerate, they can only rely on interpolation, which is actually very passive.

---

> ### Author Response · Authors · 2023-11-18
> **Response to Reviewer 9SXt (part 2)**
>
> _**Q1: How can we determine if the feet sliding shown in Figure 4 is a result of training with a low framerate?**_
>
> We thank the reviewer for this great reminder. We provide clearer explanation for qualitative visualization in Figure 4 of main paper. As baseline cannot directly generate high-framerate motions, we let it first generate conventional 20 fps (blue), and **conduct interpolation towards target 100 fps** (purple, pink, yellow, green). Floating feet in these 4 colors indicate the sliding phenomenon caused by interpolation. However, for our NeRM, we can **directly** generate 100 fps (orange+purple, pink, yellow, green) with reasonable visual effect. Note that we did not draw every frames in 100 fps as it may appears too crowded, so instead, we only draw frames at 20 fps in orange to present overall generation, with two enlarged views of 100 fps details in the figure. Similarly, Figure 5 of Sec. E.6 in Appendix is also the same. In baseline, the blue is baseline 20 fps, and other colors are interpolation towards target framerate. In NeRM, all frames are directly generated, and we draw at 20 fps as orange, with high-framerate details in other colors.
> Additionally, these figures, along with our subsection _High-framerate motion generation_, have proved that low-framerate generation with interpolation is not sufficient to generate high-quality high-framerate motions. This could also be a response for the above W1.
>
>
> _**Q2.1: Does the use of variational INR, i.e., normalizing the latent code to a normal distribution with KL loss, affect the preservation of high-framerate details?**_
>
> We appreciate the thoughtful comment. To some extent, the use of variational INR introduces a slight degradation in motion reconstruction during our first stage. However, **it is quite important for the second stage (diffusion stage)** due to a smoother representation space (explained in Q2.2). Please refer to Table 3 and related analysis in Sec. E.3 of Appendix. We also highlight this part in blue for emphasis. The table illustrates that variational INR can better preserve high-framerate details compared to non-variational INR, as evident from the clip-FID scores. Additionally, we observe that the reconstruction errors or non-variational INRs is smaller than variational ones, as non-variational INRs may overfit the motion sequence by powerful neural network. These findings suggest that the variational approach effectively regularizes the latent space and improves the sampling of new motions.
>
> _**Q2.2: Such normalization typically leads to a smoother representation space?**_
>
> Yes, such normalization does lead to a smoother representation space. In this context, each latent code **$z_{i}$** is optimized with an over-parameterized decoder $f_{\theta}$ to reconstruct a single sample. This makes the distribution of latent codes complex, and results in a representation space where semantic similarity does not correspond to a simple distance measure. Consequently, interpolations between representations in this space may lack meaningful interpretations. To avoid such a complex representation space, we normalize the latent code **$z_{i}$** to a normal distribution using KL loss. This regularization of the latent distribution ensures a more meaningful and interpretable latent space. By increasing the weight of the KL loss, the network tends to map different input samples to similar regions in the latent space. This constraint encourages the model to learn a smoother representation space where similar samples are closer to each other in the latent space.

---

> ### Comment · Reviewer_9SXt · 2023-11-22
> **Thanks for your reply.**
>
> I have no further questions.

---

### Official Review · Reviewer_VxZF · 2023-11-01

**Soundness:** 3 good
**Presentation:** 3 good
**Contribution:** 3 good
**Rating:** 8
**Confidence:** 3

**Summary:**

The paper addresses the challenge of generating realistic human motions at high framerates, a task made difficult by inconsistent training data framerates, memory constraints, and the slow performance of generative models. Current solutions downsample high-framerate details or discard low-framerate samples, leading to information loss. The authors propose NeRM, a generative model utilizing Implicit Neural Representations (INRs) to harness varied-size data for high-framerate motion synthesis without explicitly modeling raw motions. NeRM not only outperforms other methods but also efficiently produces motions at any desired framerate while remaining memory-efficient.

**Strengths:**

1. The paper proposes to use a novel variational INR to generate arbitrary framerate motion. The experimental results support this claim that NeRM outperforms other methods in different framerate generation.

2. It enables to generalize INR to the new data without retraining by introducing the latent code to the INR input.

3. The presentation is overall clear. The core idea as well as the technical details are well presented and easy to follow.

4. The performance outperforms other baselines.

**Weaknesses:**

1.  As for the generation part, the part of the input for INR is z, it seems that the model heavily depends on the quality of the latent representation z.

2. The idea of using time INR has been use in NeMF: Neural Motion Fields for Kinematic Animation, which may limit the novelty contribution of this paper.

**Questions:**

1. Two-stage training gives good performance. It would be good to give more details on training the auto-encoding of the latent code.

2. In order to avoid retraining of INR, some other methods choose to use latent code to modulate the weights of INR. Is there any insight for the choice in the paper?

3. How do you determine the number of codes in the codebook?

---

> ### Author Response · Authors · 2023-11-18
> **Response to Reviewer VxZF (part 1)**
>
> We thank the reviewer for the thoughtful comments. We are glad that the reviewer finds our method novel, the core idea easy to follow, as well as good performance. Below we address all the concerns.
>
> _**W1: It seems that the model heavily depends on the quality of latent representation $z$.**_
>
> We agree with the reviewer that the latent representation **$z$** is a crucial variable in our NeRM. It acts as a bridge between the variational INR model $f_{\theta}$ and the diffusion model $\epsilon_{\phi}$. This is similar to latent representation in Stable Diffusion [Ref1]. In our context, the quality of latent representation can be influenced by the dimension and the variational distribution $\mathcal{N}(\mu,\Sigma)$ of **$z$**. Therefore, we add ablation studies to investigate the impact of different dimensions $d$ of **$z$** and weight $\lambda_{KL}$ on the KL-loss for motion reconstruction. When the dimension $d$ is set to 256, the motion reconstruction loss becomes significantly small, and higher dimension can slightly improve the reconstruction ability. For $\lambda_{KL}$, a higher weight results in a smoother latent space but increases the reconstruction error. Following Stable Diffusion, we set $\lambda_{KL}$ as a small value (i.e. 1e-4). Overall, we find that the performance of our method is not very sensitive to both parameters in the wide range. The details concerning the above are added in Sec. E.3 of Appendix.
>
> | $d$ | 128 | 256 | 512 |
> | --- | --- | --- | --- |
> | MRE | 0.0459 | 0.0412 | **0.0407** |
>
> | $\lambda_{KL}$ | 1e-5 | 1e-4 | 1e-3 |
> | --- | --- | --- | --- |
> | MRE | **0.0410** | 0.0412 | 0.0459 |
>
>
> _**W2: The idea of using time INR has been used in NeMF, which may limit the novelty contribution.**_
>
> We respectfully disagree with the reviewer that the idea of using time INR in our method is not novel due to the presence of NeMF. Admittedly, both NeMF and our method are based on time INR. However, they are quite different with respect to the following main aspects.
> **(1) High-framerate motion generation.** The core idea of our NeRM is to generate high-framerate human motions using available yet limited motion datasets with textual annotations. These valuable datasets, however, consists of multi-framerate data, ranging from 20fps to 250fps. We therefore meticulously design an INR framework to take full advantage of such varied-size data. On the other hand, NeMF adopts a VAE framework that requires explicit modeling of raw motions for input to the VAE encoder, leading to the inability to support training at arbitrary framerate. While NeMF can roughly achieve high-framerate generation by adjusting the temporal coordinates, the quality of the generated motions is significantly poor, resembling common motion interpolation techniques.
> **(2) Conditional generation.** Although both of NeMF and NeRM support unconditional generation, our research is orthogonal to NeMF in terms of conditional generation. Particularly, NeMF supports generation with internal conditions (such as keyframes, which belong to motion itself); while NeRM is designed for generation with external condions (such as text prompts and action labels). Another difference is that the deterministic optimization process makes NeMF only generate one motion, limiting probabilistic motion synthesis; while our NeRM supports diverse motion generation.
> **(3) Generative framework.** NeMF is trained in the form of VAE, so it has the limitation of not always giving satisfactory results when sampling the latent space. Instead, our NeRM integrates the diffusion model into the neural motion representation, achieving impressive performance of motion generation.
>
> Note that the limitation (2) and (3) of NeMF can be found in Sec. 5.1 of NeMF original paper [Ref2].

---

> ### Author Response · Authors · 2023-11-18
> **Response to Reviewer VxZF (part 2)**
>
> _**Q1: It would be good to give more details on training the auto-encoding of the latent code.**_
>
> We thank the reviewer for the suggestion. It is required to emphasize that unlike a traditional **auto-encoder** whose latent code is produced by the encoder, we draw inspiration from DeepSDF [Ref3] in 3D shapes and use an **auto-decoder** to learn the latent code without an encoder. This encoder-less design avoids explicit modeling of raw motions, making arbitrary-framerate training feasible. To construct distributions where one can sample a representation to generate a new high-quality motion, we gain insight from Variational Auto-Encoder (VAE) that utilizes an encoder to infer a distribution from which representations of motions can be sampled, and employs a decoder to reconstruct the data from the representation. _As our design does not contain an encoder,_ we model each motion by sampling from a learnable distribution with optimized parameters, i.e., mean $\mu$ and covariance $\Sigma$, leading to learnable parameters ($\mu_{i}$, $\Sigma_{i}$) for $i$-th motion in training dataset. This is our variational INR framework. When training, we sample a latent representation **$z_{i}$** from a distribution $\mathcal{N}(\mu_{i},\Sigma_{i})$. We then combine **$z_{i}$** with temporal coordinate encodings $\gamma(t)$, and input them into a shared MLP decoder $f_{\theta}$. The parameters $\mu_{i}$, $\Sigma_{i}$ and $\theta$ are optimized simultaneously by minimizing the reconstruction loss between the generated and ground truth motion. The details concerning the above are added in Sec. B.2 of Appendix.
>
> _**Q2: Some other methods choose to use latent code to modulate the weights of INR. Is there any insight for the choice in the paper.**_
>
> Indeed, both (a) _modulating features (in our paper)_ and (b) _modulating weights of INR (the reviewer mentioned)_ belong to **Generalizable INRs**, which aim to learn common representations of a MLP across instances and adapt to unseen data instances. The simple and widely-used (such as SDF, NeRF) _feature-modulation_ concatenates the latent code into the features of MLP as the input condition and shares whole MLP weights across data instances; while the _weight-modulation_ learns to update the part or whole MLP weights for high performance with unstable and expensive training. Feature-modulation is a simple formulation we adopted for its significantly computational efficiency, and our experimental results show that the feature-modulation is sufficient to achieve good reconstruction of human motions. More advanced designs such as weight-modulation models for supporting arbitrary-framerate training can be left for future work.
>
> _**Q3: How do you determine the number of codes in the codebook?**_
>
> We directly use the pre-trained codebook reported in [Ref4], which is trained on the HumanML3D dataset (the same dataset as ours). [Ref4] provides an ablation on the number of codes in Sec. 4 of their supplemental material, and suggests that the performance of 512 codes is slightly better than 1,024 codes while 256 codes are not sufficient for reconstruction. We follow this work and determine the 512 codes in the codebook.
>
> **References:**
>
> [Ref1] Rombach et al. High-Resolution Image Synthesis with Latent Diffusion Models. CVPR, 2022.
>
> [Ref2] He et al. NeMF: Neural Motion Fields for Kinematic Animation. NeurIPS, 2022.
>
> [Ref3] Park et al. Deepsdf: Learning Continuous Signed Distance Functions for Shape Representation. CVPR, 2019.
>
> [Ref4] Zhang et al. Generating Human Motion From Textual Descriptions With Discrete Representations. CVPR, 2023.

---

### Author Response · Authors · 2023-11-18
**General Response to All Reviewers**

We thank the reviewers for the thorough review and insightful comments. We are happy to see reviewers recognize NeRM as a **“both interesting and innovative work for neural continuous motion representation” (9SXt)**, **“accepting arbitrary framerate while supporting high-framerate generation” (KnYZ)**, **“address the underexplored task” (WpJV)** and **“a novel variational INR, easy to follow, outperform other baselines” (VxZF)**.

In the revised version of the paper, changes are marked in **blue**. The summary of changes is as below:

**--Reviewer VxZF**

Additional ablations on different dimensions $d$ and weights $\lambda_{KL}$ **(Sec. E.3, Table 5 Appendix)**

More details on training the auto-decoder of the latent code **(Sec. B.2 Appendix)**


**--Reviewer 9SXt**

A clearer explanation of  **Figure 4 main paper**

Ablations on the effectiveness of variational INRs **(Sec. E.3, Table 3 Appendix)**


**--Reviewer KnYZ**

A clearer explanation of Eq 2 **(Sec. 3.1 main paper)**

A clearer explanation of maximum framerate limitation **(Sec. B.3, Sec. F Appendix)**

Experimental comparison between NeMF and NeRM on unconditional generation setting **(Sec. 4.3 main paper)**

Ablations of normalizing time positions **(Sec. E.3, Table 6 Appendix)**


**--Reviewer WpJV**

Discussing limitations of our work **(Sec. F Appendix)**

More related works as suggested by the reviewer **(Sec. 1 main paper, Sec. A Appendix)**

More details of clip-FID **(Sec. C.2, Figure 2 Appendix)**

We again thank you for your valuable feedback and comments. We would appreciate you being able to review our responses and let us know if you have any additional concerns or questions. And we are happy to clarify further.

---

### Meta-Review · Area_Chair_MLER · 2023-12-06

**Metareview:**

The paper introduces NeRM, a novel generative model for high-framerate human motion synthesis. NeRM utilizes Implicit Neural Representations (INRs) to learn continuous motion fields, allowing it to efficiently generate realistic motions at any desired framerate. It addresses the challenges of inconsistent training data framerates and memory constraints, outperforming existing methods by fusing data from different framerates and employing progressive training. NeRM achieves high-quality motion generation.

Overall, the reviewers are positive and mild. The strengths include addressing the underexplored task, innovative approach, various experiments, clear presentation, and good performance.
The weaknesses and concerns include advantages over NeMF (and novelty), and not convincing motivation.

Most of the reviewers' concerns and questions have been well-addressed by the authors through the rebuttal.

One negative reviewer left a remaining key concern about the novelty of the proposed method over NeMF (He et al.). However, to this AC, the point seems to be properly addressed by the authors.

**Justification For Why Not Higher Score:**

The proposed method focuses on a fairly specific application of 3D human motion, in particular, for encompassing any framerate input, which is of interest to a subset of the community.

The proposed modeling is interesting and showed effectiveness on the application to some extent. However, if the authors focused on the modeling itself and showed its effectiveness on various applications, it would have more impact.

**Justification For Why Not Lower Score:**

This work enables any framerate input which would improve the data efficiency of the related field.

---

### Decision · Program_Chairs · 2024-01-16

Accept (poster)